# Comparison of different methods to retrieve optical-equivalent snow grain size in central Antarctica

Tim Carlsen[1], Gerit Birnbaum[2], André Ehrlich[1], Johannes Freitag[2], Georg Heygster[3], Larysa Istomina[3], Sepp Kipfstuhl[2], Anaïs Orsi[4], Michael Schäfer[1], and Manfred Wendisch[1]

[1]Leipzig Institute for Meteorology, Leipzig University, Leipzig, Germany
[2]Alfred Wegener Institute, Helmholtz Centre for Polar and Marine Research, Bremerhaven, Germany
[3]Institute of Environmental Physics, University of Bremen, Bremen, Germany
[4]Laboratoire des Sciences du Climat et de l'Environnement, Gif-sur-Yvette, France

*Correspondence to:* Tim Carlsen (tim.carlsen@uni-leipzig.de)

**Abstract.** The optical-equivalent snow grain size affects the reflectivity of snow surfaces and, thus, the local surface energy budget in particular in polar regions. Therefore, the specific surface area (SSA), from which the optical snow grain size is derived, was observed for a two-month period in central Antarctica (Kohnen research station) during austral summer 2013/14. The data were retrieved on the basis of ground-based spectral surface albedo measurements collected by the COmpact RAdiation measurement System (CORAS) and airborne observations with the Spectral Modular Airborne Radiation measurement sysTem (SMART). The Snow Grain Size and Pollution amount (SGSP) algorithm, originally developed to analyze spaceborne reflectance measurements by the MODerate Resolution Imaging Spectroradiometer (MODIS), was modified in order to reduce the impact of the solar zenith angle on the retrieval results and to cover measurements in overcast conditions. Spectral ratios of surface albedo at 1280 nm and 1100 nm wavelength were used to reduce the retrieval uncertainty. The retrieval was applied to the ground-based and airborne observations and validated against optical in situ observations of SSA utilizing an IceCube device. The SSA retrieved from CORAS observations varied between $27\,\mathrm{m^2\,kg^{-1}}$ and $89\,\mathrm{m^2\,kg^{-1}}$. Snowfall events caused distinct relative maxima of the SSA which were followed by a gradual decrease in SSA due to snow metamorphism and wind-induced transport of fresh fallen ice crystals. The ability of the modified algorithm to include measurements in overcast conditions improved the data coverage especially at times when precipitation events occured and the SSA changed quickly. SSA retrieved from measurements with CORAS and MODIS agree with the in situ observations within the ranges given by the measurement uncertainties. However, SSA retrieved from the airborne SMART data slightly underestimated the ground-based results.

## 1 Introduction

The cryosphere plays a fundamental role in determining the Earth's surface radiative energy budget, as snow and sea ice represent surfaces with the highest albedo on Earth. Picard et al. (2016) estimated that a hypothetical change in global surface albedo of one percent would offset a difference in reflected energy comparable to the globally averaged radiative forcing of $1.82\,\mathrm{W\,m^{-2}}$ caused by the increase in $CO_2$ concentration since the preindustrial time (Myhre et al., 2013). This change in

global surface albedo could be caused either by a variation in snow and sea ice cover or by a change of the snow albedo itself. During boreal winter, snow and sea ice cover up to 15 % of the Earth's surface (Vaughan et al., 2013). Although these areas are mainly in polar regions with partly low values of incoming solar radiation, the high snow and sea ice albedo may facilitate substantial changes in the local surface energy budget. In Antarctica, more than 99.8 % of the continent and most of the sea ice

are covered with snow all year round (Burton-Johnson et al., 2016). However, the snow surface albedo varies both on temporal and spatial scales. For example, Munneke et al. (2008) found variations of the broadband albedo of snow in a range between 0.77 and 0.88 at five locations in Dronning Maud Land.

Changes of broadband and spectral snow albedo are caused by different parameters such as snow grain size (and shape), surface roughness (e.g., Warren et al., 1998), soot content (e.g., Bond et al., 2013), and cloudiness. It further depends on

wavelength (e.g., Hudson et al., 2006; Warren and Brandt, 2008) as well as solar position (e.g., Wiscombe and Warren, 1980; Wiscombe, 1980; Dumont et al., 2010), and varies with snow depth (e.g., Wiscombe and Warren, 1980) and liquid water content (e.g., Wiscombe and Warren, 1980; Gallet et al., 2014a). From these parameters, the snow grain size has the largest effect on snow albedo. Simulations by Wiscombe and Warren (1980) showed that the spectral albedo in the near-infrared part of the spectrum may drop by a factor of 2 or more when the snow grain size increases from $50\,\mu m$ to $1000\,\mu m$. Dang et al.

(2015) showed that the transition from fresh fallen snow with a typical snow grain size of $100\,\mu m$ to aged snow ($1000\,\mu m$) leads to a decrease in snow albedo (spectrally integrated over $0.3\text{-}4.0\,\mu m$) of 15 % from 0.83 to 0.72. This relation could clearly be identified in ground-based measurements by Domine et al. (2006) analyzing snow samples taken at Svalbard in 2001. They measured a decrease in reflectance at $1310\,nm$ wavelength by 45 % with increasing grain size from $290\,\mu m$ to $1175\,\mu m$. Libois et al. (2015) and Picard et al. (2016) retrieved a three-year time series of SSA from spectral albedo measurements at

Dome C ($75°\,6'$ S, $123°\,0'$ E), Antarctica, which emphasized the dynamical evolution of near-surface SSA by displaying a 3-fold decrease every summer in response to the increase of air temperature. A detailed comparison of this dataset with snow model simulations and a geophysical interpretation are presented by Libois et al. (2015). Similar seasonal variations but little year-to-year variation was found by Jin et al. (2008) who retrieved snow grain size from measurements with the MODerate Resolution Imaging Spectroradiometer (MODIS) at 1.64 and $0.64\,\mu m$ wavelength over the Antarctic continent for four days

each year between 2000 and 2005. Therefore, a positive feedback mechanism can be postulated: increasing snow temperatures are followed by an accelerated snow metamorphism and a decrease of the snow albedo, which leads to higher absorption and heating of the snow. However, the expected snow albedo decrease due to temperature-induced metamorphism (0.3 % for a warming of 3 K) can be overcompensated by an increase in snow albedo by 0.4 % owing to a projected increase in precipitation during the twenty-first century in interior Antarctica (Picard et al., 2012).

The larger the snow grains are, the longer is the photon path length through the individual ice crystals and the higher is the probability of photon absorption leading to a lower surface albedo (Wiscombe and Warren, 1980). This effect is most pronounced at wavelengths larger than $1000\,nm$ where the imaginary part of the complex refractive index of ice increases. Observations showed that the grain size (from visual determination) of snow crystals varies between $10\,\mu m$ for fresh fallen snow and up to $3\,mm$ for aged snow (Singh, 2001). As snow ages, the grains become larger and more spherical (Colbeck,

1983; Kaempfer and Schneebeli, 2007). Note that this snow metamorphism is also effective below the freezing temperature. It mainly depends on snow microstructure, snow temperature, and its vertical gradient within the snowpack.

The traditional grain size is defined by the length of the largest extension of a snow grain (Fierz et al., 2009; Leppänen et al., 2015), which causes ambiguities due to the complex shapes of snow grains. Therefore, not only for radiative transfer applications, the optical-equivalent grain size is introduced as the radius $r_{opt}$ of a collection of spheres with the same volume-to-surface ratio compared to the actual non-spherical snow grains (Grenfell and Warren, 1999; Neshyba et al., 2003). The specific surface area (SSA, surface area of ice-air interface per unit mass) is related to the optical radius of the snow grains and the density of ice $\rho_{ice}$ ($917\,\mathrm{kg\,m^{-3}}$) by:

$$SSA = \frac{3}{\rho_{ice} \cdot r_{opt}}. \tag{1}$$

SSA (in units of $\mathrm{m^2\,kg^{-1}}$) is measured with techniques such as methan adsorption (e.g., Domine et al., 2001; Legagneux et al., 2002), stereology (e.g., Matzl and Schneebeli, 2010), and X-ray microtomography (e.g., Flin et al., 2005; Kaempfer and Schneebeli, 2007). Those methods are difficult to employ in the field. Therefore, optical measurements that utilize the spectral absorption of snow grains are applied in field studies (e.g., Gallet et al., 2009) for the in situ measurement of SSA. However, in situ techniques to measure SSA are restricted to single observation sites. Consequently, longer time series of SSA are scarce in remote Arctic and Antarctic areas.

To retrieve SSA and $r_{opt}$, measurements of reflected solar radiation are required (e.g., Bohren and Barkstrom, 1974; Wiscombe, 1980; Grenfell et al., 1994). The retrievals are based on the spectral variability of snow albedo as a function of optical snow grain size. Snow albedo models are employed to calculate the spectral albedo and to invert the measurements to retrieve the optical snow grain size (e.g., Wiscombe and Warren, 1980). These albedo models mostly assume spherical grains, which is unrealistic because the grain shape is usually far from being spherical (e.g., Kokhanovsky and Zege, 2004; Libois et al., 2013; Leppänen et al., 2015; Krol and Löwe, 2016). Picard et al. (2009) estimated an uncertainty of $\pm 20\,\%$ when determining SSA from albedo measurements in case of an unknown snow grain shape. A common approach to account for the non-spherical shape of snow grains is to represent the non-spherical snow grains by a population of spherical grains with the same area-to-mass ratio in the spectral albedo model. However, as shown by Dang et al. (2016), this approximation can lead to an underestimation of the retrieved optical snow grain size by a factor of more than 2.

Several algorithms have been developed that consider snow grains of irregular shape (e.g., Kokhanovsky and Zege, 2004; Lyapustin et al., 2009). The Snow Grain Size and Pollution amount (SGSP) retrieval algorithm by Zege et al. (2011) to analyze satellite observations by MODIS was validated against ground-based in situ measurements from the Arctic, the Antarctic, Greenland, and Japan revealing a correlation coefficient of 0.86 (Wiebe et al., 2013). Up to now, grain size products of polar orbiting satellites do not provide a sufficiently high temporal resolution that may resolve snow precipitation and metamorphism that typically can advance in a matter of hours.

In this study, ground-based measurements with high temporal resolution were utilized to study the evolution of optical snow grain size in central Antarctica. Independent methods are introduced and applied in Sect. 2. The SGSP retrieval algorithm was advanced and adapted to ground-based spectral albedo measurements as discussed in Sect. 3. The resulting time series of

optical snow grain size estimates is presented in Sect. 4 including results from in situ measurements as well as from remote sensing by ground-based and additional airborne and satellite observations.

## 2   Measurements and instrumentation

The measurements were conducted at and in the vicinity of the Kohnen research station operated by the Alfred Wegener Institute, Helmholtz Centre for Polar and Marine Research (AWI). Kohnen station is located at the outer part of the East Antarctic plateau (75° 0' S, 0° 4' E, 2892 m above sea level), approximately 500 km from the coastline, where the local weather is mostly determined by weak catabatic winds. The annual snow accumulation is 62 mm liquid water equivalent (Oerter et al., 2000) with moderate snowfall (1 mm to more than 5 mm water equivalent) occuring only a few times per year (Birnbaum et al., 2006). The atmosphere is both clean with an Aerosol Optical Depth (AOD) at Kohnen station of 0.015 (at 500 nm, measured 2001-2006, Tomasi et al., 2007) and dry (mean integrated atmospheric water vapor between December 2013 and January 2014: $1.1 \, \mathrm{kg \, m^{-2}}$). The black carbon (BC) concentration in the snowpack on the Antarctic plateau is low; for the South Pole, Hansen and Nazarenko (2004) reported a mean BC concentration of 0.2 ppbw (parts per billion by weight).

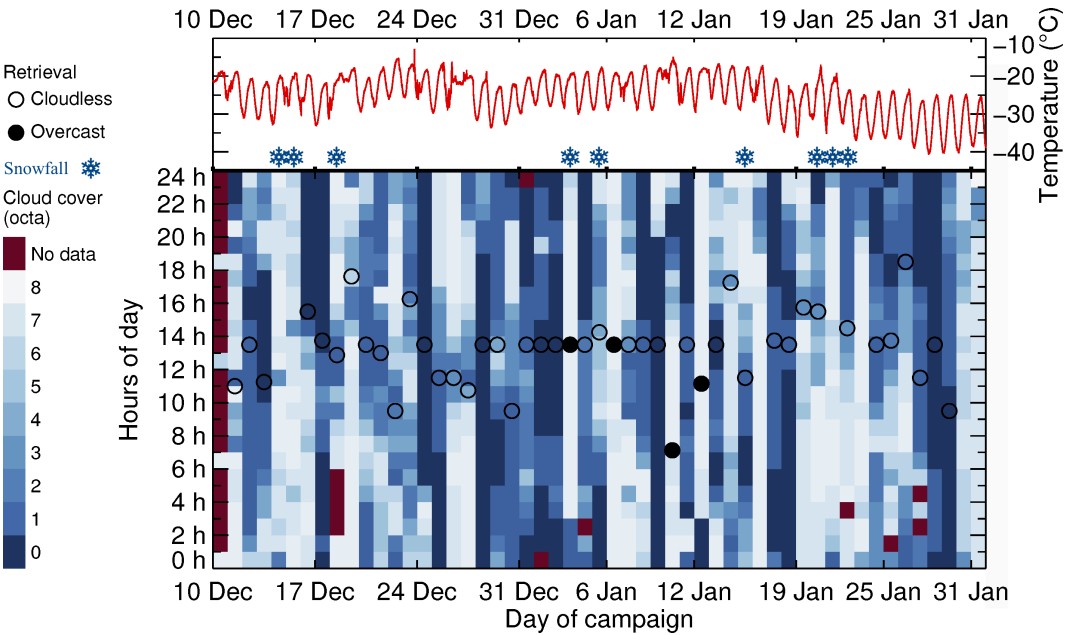

**Figure 1.** Time series of 2 m air temperature (red line) and hourly cloud cover (blueish squares) at Kohnen station between 10 December 2013 and 31 January 2014. Snowflake symbols denote days with snowfall. Black circles show times when SSA was retrieved from CORAS measurements (Fig. 6) and denote retrievals for cloudless (open circles) and overcast (filled circles) conditions.

## 2.1 Ground-based observations

The ground-based measurements were embedded in the 'Coldest Firn' (CoFi) project by AWI, which investigates microstructure as well as physical and chemical composition of snow and firn on the Antarctic plateau. Solar broadband and spectral radiation measurements of snow surface albedo combined with meteorological observations (temperature, humidity, wind velocity and direction, radio sounding, synoptic observations) were conducted. In addition, optical snow grain size and density of snow samples were measured in situ on a daily basis at Kohnen station, accompanied by measurements of vertical snow temperature profiles. Snow samples from in situ measurements were extracted daily (except on 5 December, 12, 17, 18, and 19 January) along a 100 m-transect. After extraction, the measurements of SSA were conducted with an IceCube device, which uses a laser diode at 1310 nm to illuminate the snow sample underneath an integrating sphere (Gallet et al., 2009). The reflected signal is detected by a photodiode. By means of a certified standard, the hemispherical infrared reflectance is derived, which is used to calculate SSA and $r_{\mathrm{opt}}$ applying a radiative transfer model (for details, see Gallet et al., 2009). For the derived SSA values between 5 to $130 \, \mathrm{m^2 \, kg^{-1}}$, the measurement uncertainty is in the range of 10 %. SSA values above $60 \, \mathrm{m^2 \, kg^{-1}}$ require caution as the insufficient optical depth of the snow sample may cause artifacts as reported by Gallet et al. (2009). However, the densities of the snow samples for which Gallet et al. (2009) reported this limitation were below $100 \, \mathrm{kg \, m^{-3}}$, whereas the observed snow densities within this study were all well above this value (around 60 % of the samples with densities between 280-350 $\mathrm{kg \, m^{-3}}$). The higher optical depth of the samples might indicate a higher limit for the SSA measurements. However, SSA values above $60 \, \mathrm{m^2 \, kg^{-1}}$ occured only in 10 % of the measurements.

Two downward-looking digital cameras were employed for ground-based photogrammetric measurements of surface roughness structures. Another digital camera was used to resolve the hemispherical-directional reflectance factor (HDRF) of the snow surface following a method described by Ehrlich et al. (2012). For the definition of reflectance quantities used within this study, we refer to Schaepman-Strub et al. (2006). Furthermore, an all-sky camera was used for cloud observations. AOD was determined by a sun photometer.

The ground-based measurements were carried out during austral summer between 4 December 2013 and 31 January 2014. Within this period, nine snowfall events occured (recorded by visual observation), surface temperature ranged from -40 °C to -15 °C. The time series of the measured 2 m air temperature and the hourly cloud fraction are presented in Fig. 1. The temperature indicates diurnal cycles on most days. Only on days with high cloud cover almost constant air temperatures were observed. Towards the end of the measurement period, the temperature level decreased due to lower sun elevations. The mean total cloud amount over the entire period was less than 4 octa. Only five completely overcast days were reported. The total cloud cover was highly variable and mainly influenced by cirrus clouds.

The spectral snow albedo $\alpha(\lambda)$ was measured from ground-based and airborne instruments. At Kohnen station, the COmpact RAdiation measurement System (CORAS) measured upward and downward spectral irradiance $[F^{\uparrow}(\lambda), F^{\downarrow}(\lambda)]$ within 0.3 to 2.2 μm wavelength. The spectral snow albedo was obtained by:

$$\alpha(\lambda) = \frac{F^{\uparrow}(\lambda)}{F^{\downarrow}(\lambda)}. \tag{2}$$

The spectral resolution is 2 to 3 nm between 0.3 and 1.0 µm and 15 nm up to 2.2 µm wavelength, with a full spectrum measured every 15 s. The uncertainties of albedo measurements with CORAS range between 4.0 to 8.0 % depending on wavelength and combining different sources of instrumental errors. The signal-to-noise ratio accounts for 1.3-3.0 % uncertainty. Dark spectra were recorded constantly throughout the measurements resulting in a reliable correction for dark current and stray light within the spectrometer (0.1 % uncertainty). The wavelength calibration of the spectrometer accounts for 1.0 % uncertainty. For albedo measurements, the two optical inlets were cross-calibrated with an identical radiation source at four times during the observation period. The temporal stability of this cross-calibration is estimated with 1.0-4.5 % depending on wavelength. Furthermore, the non-ideal cosine characteristics of the irradiance optical inlets were characterized within the laboratory. The optical inlets were mounted on a turn table and the lamp signal under different angles of incidence (-95° to +95° in steps of 5°) was recorded. This procedure was repeated for four different relative azimuth angles between lamp and optical inlet and was used to compute correction factors for the cosine response depending on solar zenith angle, wavelength, and the direct fraction of the global irradiance. The azimuthal stability of the correction factors of 3.5 % was used to estimate the instrumental errors attributed to the non-ideal cosine response of the optical inlets. This way, the instrumental uncertainties combine to 6.8 % at 1280 nm wavelength. The analyzed measurement times were carefully selected to avoid errors due to frost formation and shadow effects which were typically observed during early morning.

The location and mounting of the ground-based instrumentation are illustrated in Fig. 2. Table 1 lists the ground-based and airborne instruments relevant to investigate the evolution of snow microphysical and optical properties.

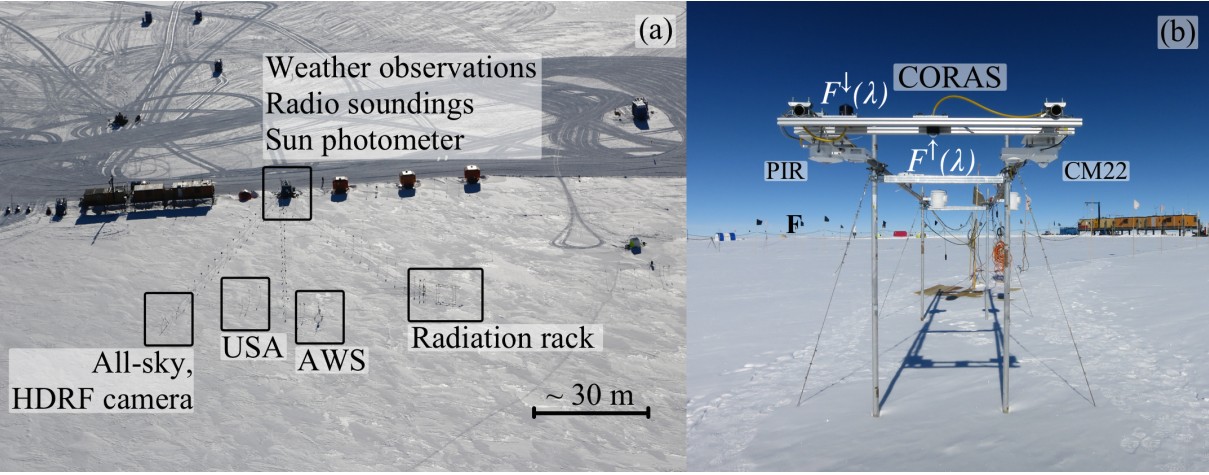

**Figure 2.** Left (a): Aerial photograph of Kohnen station. The positions of different ground-based instruments are marked with black rectangles. USA: Ultrasonic anemometer. AWS: Automatic weather station. Right (b): Instrument setup at the ground-based radiation rack: optical inlets of the CORAS instrument and the broadband radiation instruments CM22 and PIR.

## 2.2 Airborne data

To characterize the representativeness of the local ground-based observations, an intensive observation phase including airborne measurements using the Polar 6 research aircraft from AWI was conducted between 24 December 2013 and 5 January 2014. The aircraft flew 60 hours to characterize the spatial variability of snow properties above Dronning Maud Land. A map of the tracks of the 18 research flights is shown in Fig. 3. During each flight, an overpass over Kohnen station was realized to compare airborne and ground-based measurements. The airborne observations covered a wide variety of surface roughness structures as well as precipitation patterns, which strongly influence snow albedo. Beside solar broadband and spectral radiation measurements, the airborne observations included measurements of snow HDRF by means of a digital camera and surface roughness measurements using a laser scanner. Meteorological measurements were provided by the Aircraft-Integrated Meteorological Measurement System (AIMMS20). The aircraft instrumentation additionally included geophysical observations within the CoFi project (snow and ice thickness).

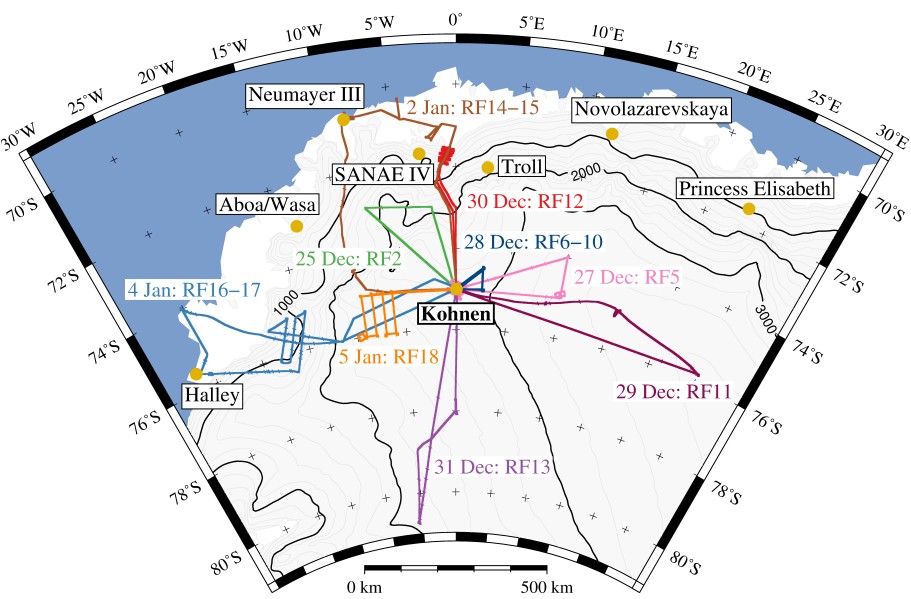

**Figure 3.** Map of flight patterns with the Polar 6 research aircraft during the campaign. Each color corresponds to a different flight.

On Polar 6, irradiance measurements similar to the ground-based observations were conducted using the Spectral Modular Airborne Radiation measurement sysTem (SMART) which applies an active horizontal stabilization of the optical inlets to correct for aircraft movement (Wendisch et al., 2001; Ehrlich et al., 2008). SMART albedo measurements have an estimated uncertainty between 4.1 to 8.1 % also taking into consideration uncertainties due to the active horizontal stabilization of 1.0 %.

## 3   Methodology

### 3.1   SGSP retrieval algorithm using satellite data

Absorption of solar radiation by ice becomes crucial in the near-infrared part of the solar wavelength spectrum (Warren and Brandt, 2008). As the effective photon path is longer within larger snow grains, the magnitude of absorption is determined by the size of the snow grains. Hence, snow surface albedo in the near-infrared mainly depends on the optical snow grain size. Zege et al. (1991) showed that the plane albedo $\alpha_\mathrm{p}(\theta_0)$ as a function of solar zenith angle $\theta_0$ can be parameterized using an asymptotic solution of the radiative transfer theory by:

$$\alpha_\mathrm{p}(\theta_0) = \exp\left[-y \cdot K_0(\theta_0)\right], \qquad \text{with} \tag{3}$$

$$K_0(\theta_0) = \frac{3}{7}\left(1 + 2\cos\theta_0\right). \tag{4}$$

The escape function $K_0(\theta_0)$ (dimensionless) describes the angular distribution of the number of photons escaping from a non-absorbing, semi-infinite medium. The uncertainty of $\alpha_\mathrm{p}$ introduced by the approximation of $K_0(\theta_0)$ given in Eq. 4 (Kokhanovsky and Zege, 2004) is below 2 % for $\theta_0 < 78°$. Following Kokhanovsky and Zege (2004), the parameter $y$ depends on the volumetric extinction and absorption coefficients of snow $b_\mathrm{ext}$ and $b_\mathrm{abs}$ (both in $\mathrm{m}^{-1}$) and the dimensionless asymmetry parameter (average cosine over the phase function) $g(\xi)$ of the snow grain which shape is represented by the parameter $\xi$:

$$y = 4\sqrt{\frac{b_\mathrm{abs}}{3\,b_\mathrm{ext} \cdot [1 - g(\xi)]}}. \tag{5}$$

Assuming pure snow and applying the principles of geometrical optics, the volumetric extinction and absorption coefficients of snow can be derived by (Kokhanovsky and Zege, 2004):

$$b_\mathrm{ext} = \frac{1.5\,C_\mathrm{v}}{r_\mathrm{opt}}, \qquad \text{and} \tag{6}$$

$$b_\mathrm{abs} = B(\xi) \cdot b_\mathrm{abs,ice} \cdot C_\mathrm{v} = B(\xi) \cdot \frac{4\pi\chi}{\lambda} \cdot C_\mathrm{v}. \tag{7}$$

$b_\mathrm{ext}$ and $b_\mathrm{abs}$ only depend on the volumetric concentration of snow grains $C_\mathrm{v}$ (dimensionless, $\rho_\mathrm{snow}/\rho_\mathrm{ice}$), the optical snow grain size $r_\mathrm{opt}$, the absorption enhancement parameter $B(\xi)$ (dimensionless), and the absorption coefficient of pure ice $b_\mathrm{abs,ice}$ ($\mathrm{m}^{-1}$), which is determined by the imaginary part of the complex refractive index of ice $\chi$ at wavelength $\lambda$. Consequently, Eq. 5 reduces to:

$$y = A \cdot \sqrt{\frac{4\pi\chi}{\lambda} \cdot r_\mathrm{opt}}, \qquad \text{with} \tag{8}$$

$$A = \frac{4}{3}\sqrt{\frac{2\,B(\xi)}{1 - g(\xi)}}. \tag{9}$$

The form factor $A$ accounts for the snow grain shape by merging $B(\xi)$ and $g(\xi)$ into a single parameter. Equation 8 forms the basis of the SGSP retrieval algorithm by Zege et al. (2011) who applied the $\chi$ data base by Warren and Brandt (2008)

and assumed a form factor of $A = 5.8$. The factor $A$ varies in general between 5.1 (Kokhanovsky and Macke, 1997) for fractals and 6.5 for spheres. Within the SGSP algorithm, a value of 5.8 is used as an average value over a mixture of randomly oriented hexagonal plates and columns with $B(\xi) \approx 1.5$ and $g(\xi) \approx 0.84$. This is in accordance with Libois et al. (2014) who recommend a value for $B(\xi)$ within $1.6 \pm 0.2$ based on measurements in Antarctica and the French Alps. Any inaccuracy in the form factor $A$ will affect the snow grain size retrieval in addition to instrumental errors. According to Zege et al. (2011), the retrieval inaccuracy which may stem from a false assumption of $A$ is less than 25 %. For the retrieval of optical snow grain size from satellite data, snow-atmosphere radiative interactions have to be taken into account by employing an atmosphere model as described in Zege et al. (2011). The effect of the atmosphere has been removed employing the radiative transfer model RAY (Tynes et al., 2001) using the subarctic winter atmospheric model (Kneizys et al., 1996) and the Arctic background aerosol model (Tomasi et al., 2007) for constant atmospheric conditions. The effect of the very low pollution in Antarctica (e.g., AOD of 0.015 at 500 nm at Kohnen station, Tomasi et al., 2007) onto the retrieval is considered to be negligible at the retrieval channels.

Radiance data from MODIS (Level 1B Collection 5) onboard the Aqua and Terra satellites were used to retrieve optical snow grain sizes in the area covered by the campaign. The SGSP algorithm was applied for areas identified as cloudless. After a preliminary separation of snow pixels, the optical snow grain size of each pixel is retrieved from radiance measurements of MODIS channels 3 (469 nm wavelength), 2 (858 nm), and 5 (1240 nm). Assuming a spectrally constant bidirectional reflectance distribution function (BRDF) of snow and using the combination of three spectral channels, the angular dependency of the measured radiance is excluded.

The final product of estimated optical snow grain sizes is provided in 2D-maps of spatial resolution of 1 km. For the local optical snow grain size at Kohnen station, the spatial average of the 50 x 50 pixel of MODIS surrounding the geographic coordinates of Kohnen was calculated. Daily averages combine up to four MODIS overpasses per day under cloudless conditions. For the solar zenith angle range between 60° and 80°, the relative error of the retrieval is below 10 % for optical snow grain sizes between 30 µm to 300 µm. It grows with increasing solar zenith angle and gets as high as 20 % for $\theta_0 = 85°$, and even higher for lower sun elevations. Therefore, the retrieval is generally not applied for $\theta_0 > 85°$. The large uncertainty of the SGSP retrieval for high solar zenith angles is related to the assumptions on the particle form factor $A$ and the approximation of the escape function $K_0(\theta_0)$ which is less accurate for oblique illumination angles. In combination with the strong forward scattering characteristic for snow grains, small errors in the assumed $A$ and $K_0(\theta_0)$ can greatly distort the albedo. Similarly, the spectral behavior of the BRDF of snow slightly depends on the illumination angle (Zege et al., 2011). Consequently, within this work the satellite retrieval is limited to $\theta_0 \leq 75°$. Furthermore, an additional uncertainty of 2 % originates from the atmospheric model.

### 3.2 Retrieval of optical snow grain size from spectral albedo measurements

In contrast to the satellite observations, for the ground-based and airborne spectral albedo measurements, the atmospheric influence can be neglected because of the high surface elevation providing a dry and aerosol-free atmosphere (Wendisch et al., 2004). To test this assumption, the direct fraction of global irradiance was simulated with the library for radiative transfer

libRadtran by Mayer and Kylling (2005) using the discrete ordinate radiative transfer solver DISORT by Stamnes et al. (1988). The radiosondes released up to four times a day were used for meteorological input (profiles of air temperature, relative humidity, and static air pressure). The contribution of direct solar radiation to the global irradiance measured at the surface was estimated to vary between 94.6 and 99.8 % at the wavelengths used in the retrieval algorithm. Therefore, the simulated diffuse part hardly exceeded 5 % of the total incident irradiance. Hence, after careful selection of cloudless periods, the optical snow grain size can be calculated directly by inverting Eqs. 3 and 8:

$$r_{\mathrm{opt}} = \left[ \frac{\ln \alpha_{\mathrm{p}}}{A \cdot K_0(\theta_0) \cdot \sqrt{\frac{4\pi \chi}{\lambda}}} \right]^2 .$$ (10)

The uncertainty of the retrieved $r_{\mathrm{opt}}$ is related to the measured albedo and the assumed particle shape. Especially surface albedo values close to unity are uncertain due to the small differences between upward and downward irradiance. To minimize uncertainties, the retrieval algorithm was adapted to spectral ratio measurements as introduced by Werner et al. (2013) and Brückner et al. (2014). Using ratios of measured snow albedo at different wavelengths decreases the retrieval error as the impact of measurement uncertainties is reduced. For the ground-based CORAS and airborne SMART measurements, the ratio $\mathcal{R}$ of albedo measurements at 1280 nm normalized by the albedo at a weakly absorbing wavelength of 1100 nm was used:

$$\mathcal{R} = \frac{\alpha(\lambda_1 = 1280\,\mathrm{nm})}{\alpha(\lambda_2 = 1100\,\mathrm{nm})}.$$ (11)

Equation 10 thus changes to:

$$r_{\mathrm{opt}} = \left\{ \frac{\ln \mathcal{R}}{A \cdot K_0(\theta_0) \cdot \left[ \sqrt{\frac{4\pi \chi(\lambda_2)}{\lambda_2}} - \sqrt{\frac{4\pi \chi(\lambda_1)}{\lambda_1}} \right]} \right\}^2 ,$$ (12)

with $\lambda_1 = 1280\,\mathrm{nm}$ and $\lambda_2 = 1100\,\mathrm{nm}$.

Figure 4a shows simulated snow surface plane albedos based on Eqs. 3 and 8 for the wavelength range between 300 nm and 2200 nm for different optical snow grain sizes (10 μm to 200 μm) at 54° solar zenith angle ($A = 5.8$, $\chi$-data from Warren and Brandt, 2008). In addition, it shows the MODIS channels used within the SGSP algorithm (M3, M2, and M5) and the spectral albedos $\alpha_{\mathrm{p}}$ at wavelengths $\lambda_1$ and $\lambda_2$ used for calculating the albedo ratio $\mathcal{R}$. The spectra of $\alpha_{\mathrm{p}}$ are related to the wavelength-dependence of the imaginary part of the complex refractive index of ice. Band M5 and $\lambda_1$ are situated within a spectral albedo region which is more sensitive to optical snow grain size due to stronger absorption by ice. Furthermore, a spectral albedo between 700 nm to 1600 nm wavelength measured by CORAS on 24 December 2013 (14 UTC) is shown. The data gap between 1300 nm and 1400 nm wavelength is due to low signal-to-noise ratios at these wavelengths. The retrieval principle is illustrated in Fig. 4b. For four different solar zenith angles ($\theta_0 = 50° - 80°$), it shows the dependence of the measured albedo ratio $\mathcal{R}$ with respect to the optical snow grain size utilizing Eq. 12 ($A = 5.8$, $\chi$-data from Warren and Brandt, 2008). Photons entering the snowpack under grazing angles have a higher probability of escaping the snowpack due to the pronounced forward scattering of ice crystals. This increases the spectral albedo mostly in the spectral range of stronger ice absorption.

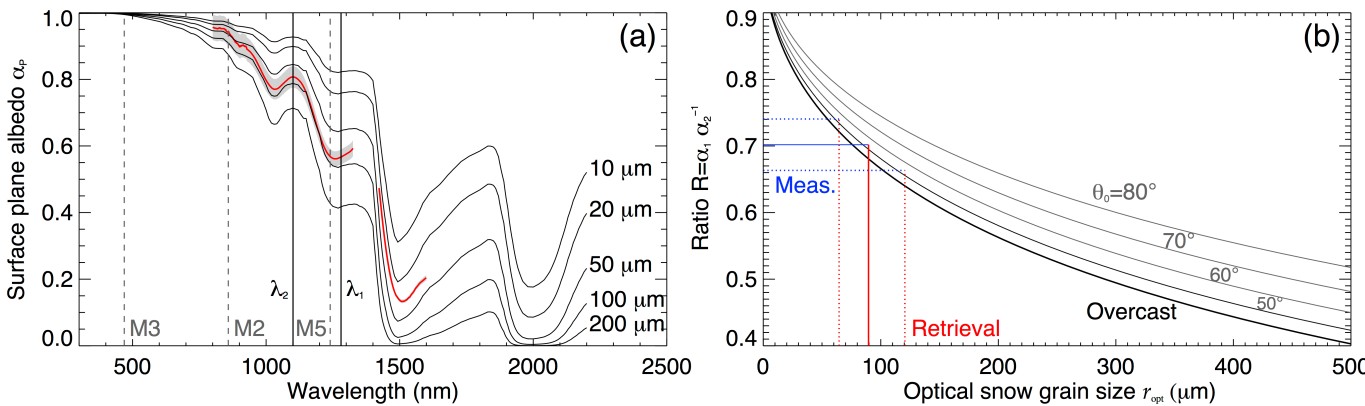

**Figure 4.** Left (a): Surface plane albedo for optical snow grain sizes between $10\,\mu m$ and $200\,\mu m$, $\theta_0 = 54°$, $A = 5.8$, $\chi$-data from Warren and Brandt (2008). M3, M2, and M5 mark MODIS spectral bands used within the SGSP algorithm. $\lambda_1$ and $\lambda_2$ denote wavelengths used within the CORAS and SMART grain size retrieval. The red solid line and shaded gray show a spectral albedo measured by CORAS on 24 December 2013 (14 UTC). Right (b): Illustration of retrieval principle. Dependence of ratio $\mathcal{R}$ with respect to optical snow grain size for different solar zenith angles (50° to 80°) and for overcast conditions, $A = 5.8$, $\chi$-data from Warren and Brandt (2008). Blue and red lines illustrate the retrieval of optical snow grain size from a measured albedo ratio $\mathcal{R} = 0.7$ with a relative uncertainty of 5.5 %.

Therefore, $\mathcal{R}$ increases with lower sun position. The overcast line in Fig. 4b corresponds to a solar zenith angle of around 50° in cloudless conditions, which is in accordance with Wiscombe and Warren (1980). The measurement uncertainty of albedo measurements at $1280\,nm$ wavelength was estimated with 6.8 %. Using $\mathcal{R}$, the uncertainty is reduced to 5.5 % as the transition to relative measurements yields independence from the cross-calibration. In addition to the instrumental errors, the surface

5  slope at the footprint scale may influence the retrieval results. Using data from Picard et al. (2016), Dumont et al. (2017) found variations of the surface slope caused by wind drift at Dome C of $\pm\,2°$. They estimated a resulting uncertainty of 10 % in retrieved SSA due to these variations of the surface slope. Assuming a similar variability of the slope of the surface at Kohnen, an additional uncertainty of 10 % is assumed for the retrieval of SSA. Applying Eq. 12 to the spectral albedo measured by CORAS on 24 December 2013 (Fig. 4a), the measured ratio $\mathcal{R}$ of $0.702\pm0.039$ leads to an estimated optical snow grain size of

10  about $90\pm31\,\mu m$ at $\theta_0 = 54°$ (blue and red lines in Fig. 4b). However, the relative uncertainty of the retrieval varies with solar position and optical snow grain size. In general, it is higher for smaller snow grains. Overall, the retrieval uncertainty ranges between 25 % and 37 % for the SSA throughout the measurement period.

### 3.2.1 Application in cloudless conditions

The retrieval algorithm was tested for specific measurements collected on a day with prevailing cloudless conditions. During

15  this day, changes in optical snow grain size are expected to be small as the last snowfall took place six days earlier. Figure 5 shows the results retrieved from ground-based measurements with CORAS (SSA and optical snow grain size) on 24 December 2013 between 6 UTC and 18 UTC. During this time, the solar zenith angle varied between 52° and 67°. Optical snow grain

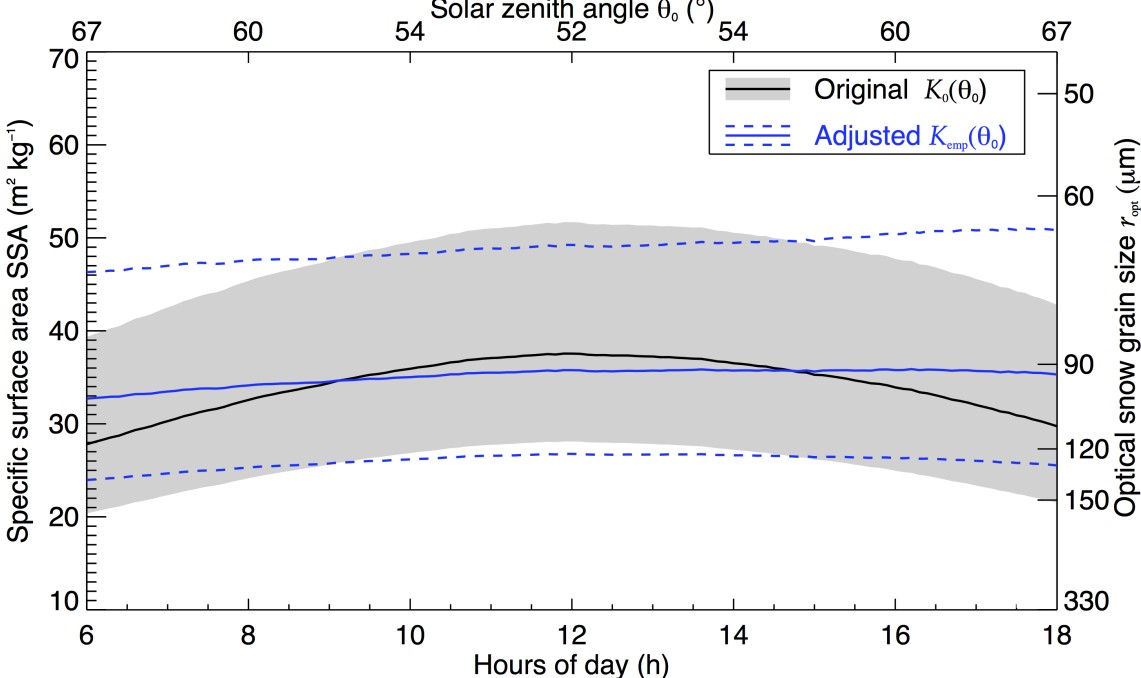

**Figure 5.** Diurnal cycle of SSA retrieved from CORAS measurements and optical snow grain size at Kohnen station on 24 December 2013. Black: SSA retrieved with original escape function $K_0(\theta_0)$ (solid line) with retrieval uncertainty (shaded gray), Blue: SSA retrieved with empirically adjusted escape function $K_{\mathrm{emp}}(\theta_0)$ (solid line) and retrieval uncertainty (dashed lines).

sizes were calculated according to Eq. 12 using the escape function $K_0(\theta_0)$ given by Eq. 4 (black line). The retrieved SSA shows a pronounced diurnal cycle and varies between 28-38 $\mathrm{m^2\,kg^{-1}}$ ($r_{\mathrm{opt}} = 86 - 117\,\mu m$). As no snowfall occured on that day, the diurnal cycle is very likely an artefact originating from the change in solar zenith angle and the assumed escape function $K_0(\theta_0)$. The approximation for the escape function (Eq. 4) might be incorrect for snow at oblique incidence due to its elongated phase function. At the same time, a representative form factor $A$ is required to account for the non-spherical snow grain shape. In order to eliminate the unrealistic diurnal cycle in the retrieved time series, the escape function was empirically adjusted to:

$$K_{\mathrm{emp}}(\theta_0) = \frac{3}{7}\,(1.5 + 1.1\,\cos\theta_0). \tag{13}$$

Applying $K_{\mathrm{emp}}$, the diurnal cycle in the retrieved SSA was significantly reduced (blue line in Fig. 5). The SSA ranges only between 33-36 $\mathrm{m^2\,kg^{-1}}$ (91-99 $\mu m$). The analysis of measurements during other cloudless days showed similar features. Therefore, $K_{\mathrm{emp}}$ was applied for the entire period of measurements. However, it should be noted that the escape function $K_{\mathrm{emp}}$ from Eq. 13 is an empirical approximation for the cases investigated here. To derive a more general description of the escape function, more cases and explicit BRDF measurements are needed, which is beyond the scope of this study. Furthermore, Gallet et al. (2014b) observed SSA variations on a sub-daily scale using the DUFISSS instrument (DUal Frequency Integrating Sphere

for Snow SSA measurements; Gallet et al., 2009) at Dome C in January 2009. They measured a drop in SSA at around noon from $40\,\mathrm{m^2\,kg^{-1}}$ to $33\,\mathrm{m^2\,kg^{-1}}$ before the SSA increased again to $41\,\mathrm{m^2\,kg^{-1}}$ at midnight. These temporal changes were attributed to the growth of sublimation crystals during daytime and nighttime formation of surface hoar. The SSA variations observed by CORAS on 24 December 2013 are in the same order of magnitude. However, the variations observed by Gallet

et al. (2014b) are not symmetric to noon and their sign changes on a day-to-day scale due to the strong dependence on meteorological conditions. Even though an influence of crystal growth processes cannot be ruled out for the measurements presented here, the evident dependence on the solar zenith angle and the constant symmetry to noon of the diurnal cycle observed in the SSA retrieved from CORAS measurements strongly indicate a dominating influence of the solar zenith angle.

    To minimize the impact of solar zenith angle even further, measurements between 13 UTC and 14 UTC were preferably

analyzed to represent the typical daily value of SSA and $r_{\mathrm{opt}}$. Using these times of day additionally ensures higher upward and downward irradiances and, therefore, a reduced measurement uncertainty by enhanced signal-to-noise ratios. The retrieval time period is also close to the probing of in situ SSA between 15 and 18 UTC which is favorable for the comparison. For each day, the times when SSA was retrieved from measurements with CORAS are added to Fig. 1 as black open circles. It has to be noted that the cloud cover was estimated from visual observation every full hour, whereas CORAS measurements

were partly analyzed for times in between the visual observation and the cloud cover given here might not be representative for the actual CORAS measurement (e.g., 5 octa on 27 December 2013). Therefore, the CORAS measurements were carefully screened for any cloud contamination by analyzing the downward solar irradiance to guarantee homogeneous cloudless or overcast conditions.

### 3.2.2   Extension to overcast conditions

The retrieval in overcast conditions was only applied when no cloudless period occured during the day. Even though new approaches to retrieve optical snow grain size below liquid clouds from airborne remote sensing are available and potentially transferable to spaceborne observations (Ehrlich et al., 2017), the SGSP algorithm is restricted to cloud-free scenes. However, in case of ground-based measurements, the analysis can be extended to cloudy conditions by using the spherical albedo and assuming isotropic illumination by the clouds. In this case, the spherical albedo $\alpha_{\mathrm{s}}$ can be expressed by using $K_0(\theta_0) = 1$

(isotropic) in Eq. 3 (Kokhanovsky and Zege, 2004):

$$\alpha_{\mathrm{s}} = \exp\left(-y\right). \tag{14}$$

Using the albedo ratio $\mathcal{R}$ in Eq. 12, the retrieved optical snow grain size is obtained in overcast conditions:

$$r_{\mathrm{opt}}^{\mathrm{cld}} = \left\{ \frac{\ln \mathcal{R}}{A \cdot \left[ \sqrt{\frac{4\pi\,\chi(\lambda_2)}{\lambda_2}} - \sqrt{\frac{4\pi\,\chi(\lambda_1)}{\lambda_1}} \right]} \right\}^2. \tag{15}$$

Equation 15 is illustrated within Fig. 4b for comparison (overcast line). Retrievals in overcast conditions were applied on four

days and are denoted as black filled circles in Fig. 1.

### 3.2.3 Retrieval using airborne measurements

For the airborne observations, SSA and $r_{\mathrm{opt}}$ were retrieved in a similar way as described above for ground-based data but using measurements of SMART. For comparison with the ground-based observations, any flight leg over the 5 x 5 pixel of MODIS surrounding Kohnen station (5 x 5 $\mathrm{km}^2$) is regarded as an overflight. The retrieval was not applied to each single measurement point but to the mean albedo measured within this box. The uncertainties of the retrieval were estimated similarly to the ground-based CORAS measurements, with the exception that the uncertainty of irradiance measurements is assumed to be slightly higher due to the remaining uncertainty in the horizontal leveling of the airborne sensors by the horizontal stabilization of SMART (6.9 % at 1280 nm wavelength). As a result, the estimated uncertainty of the measured albedo ratio $\mathcal{R}$ is about 5.6 %.

### 3.3 Influence of wavelength choice

The in situ measurements and all retrievals (in the original and adapted SGSP algorithm) use different wavelengths. Therefore, each instrument retrieves the optical snow grain size from a different depth within the snowpack. All retrievals considered in this study were performed neglecting snow layer stratification. Vertical differences in snow grain size can impose systematic differences in the retrieved values between the various instruments. To quantify the impact of the choice of wavelength, the e-folding depth $\epsilon(\lambda)$ was calculated, which is defined as the distance in the snowpack over which the irradiance reduces to $1/e$ or 37 % of the incident value and is wavelength-dependent. Following Zege et al. (1991), it is calculated by:

$$\epsilon(\lambda) = \left\{ 3 \frac{\rho_{\mathrm{snow}}}{\rho_{\mathrm{ice}}} \cdot \sqrt{2\pi \cdot \frac{\chi}{\lambda r_{\mathrm{opt}}} \cdot B(\xi) \cdot [1 - g(\xi)]} \right\}^{-1}. \tag{16}$$

The calculation assumed a mixture of hexagonal plates and columns ($A = 5.8$). The IceCube system penetrates 0.14-0.31 cm into the snowpack at 1310 nm (at snow densities between 280 and 360 $\mathrm{kg\,m}^{-3}$). At the wavelengths more sensitive to ice absorption, CORAS (SMART) mainly measures photons reflected in a depth up to 0.30 cm (at 1280 nm) and MODIS channel 5 (1240 nm) receives reflected radiation from a depth similar to the IceCube system. Hence, the penetration depth is almost identical for all measurement devices which allows a comparison of the different SSA retrievals.

### 4 Results and discussion

Figure 6 shows the time series of SSA and respective optical snow grain size derived from satellite (MODIS, red), ground-based (CORAS, blue) and airborne (SMART, green) observations between 10 December 2013 and 31 January 2014 at Kohnen station. For comparison, the in situ data are shown in black.

### 4.1 SSA and optical snow grain size from CORAS (ground-based)

SSA retrieved from CORAS measurements (blue circles in Fig. 6) varied between 27 $\mathrm{m}^2\,\mathrm{kg}^{-1}$ and 89 $\mathrm{m}^2\,\mathrm{kg}^{-1}$ throughout the campaign. The evolution of the time series revealed four pronounced maxima (minima in $r_{\mathrm{opt}}$) on 18 December 2013, 3, 17,

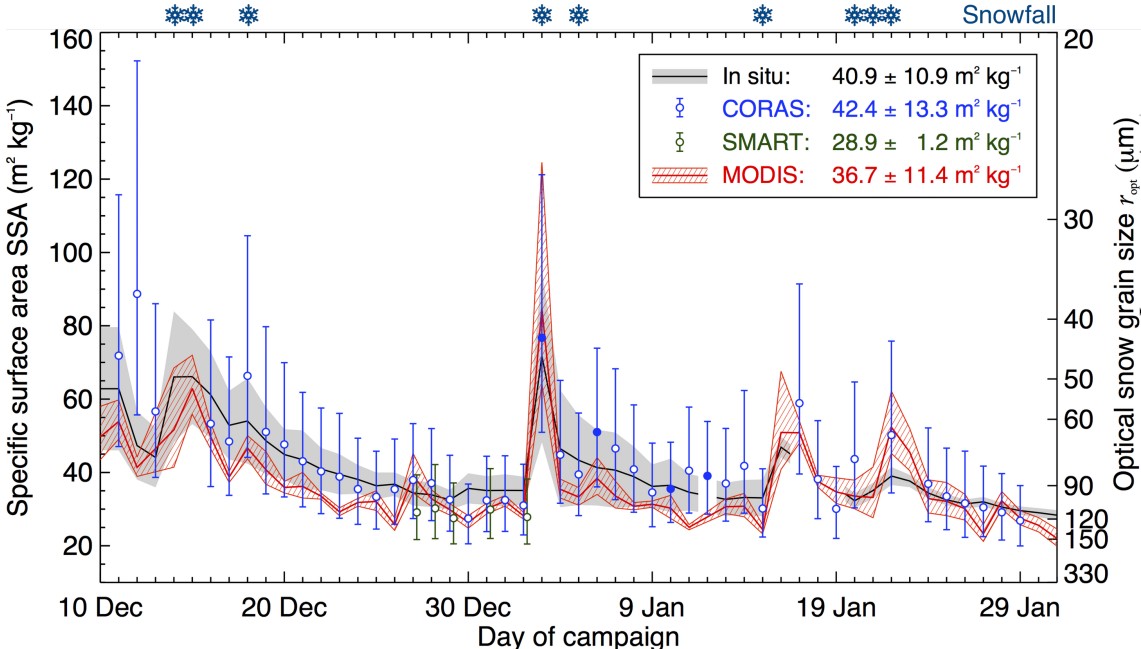

**Figure 6.** Time series of measured SSA and $r_{\mathrm{opt}}$ for the entire campaign at Kohnen station. Black: Mean (solid line) and standard deviation (shaded gray) for in situ measurements, Red: MODIS retrieval, Blue: CORAS retrieval in cloudless (open circles) and overcast (filled circles) conditions, Green: SMART retrieval in cloudless (open circles) conditions. For better visibility, the SMART data points are slightly shifted to the right on the respective dates. Top right: mean and standard deviation of the SSA retrieved by the different instruments. Snowflake symbols denote snowfall events.

and 22 January 2014, which can be related to a depth of snowfall of about 1 mm at Kohnen (visual estimation). The fresh fallen snow consists of smaller grains which increases SSA of the uppermost snow layer. The average snowfall-induced increase in SSA is $28\,\mathrm{m^2\,kg^{-1}}$. These highest SSA values are followed by a gradual decrease in SSA. From 18 to 30 December 2013, the SSA decreased daily by approximately $3.2\,\mathrm{m^2\,kg^{-1}\,d^{-1}}$ from $66\,\mathrm{m^2\,kg^{-1}}$ to $27\,\mathrm{m^2\,kg^{-1}}$. This corresponds to an increase in

5    optical snow grain size by $5.8\,\mathrm{\mu m}$ per day. This decrease in SSA is slightly slower than measured by Libois et al. (2015) at Dome C, Antarctica. They observed a drop in SSA from $90\,\mathrm{m^2\,kg^{-1}}$ to $30\,\mathrm{m^2\,kg^{-1}}$ within 10 days due to snow metamorphism. However, some abrupt decreases in SSA such as from 3 to 4 January 2014 cannot be explained by metamorphism alone, especially under the cold conditions on the Antarctic plateau. Instead, this strong increase in optical snow grain size within one day is supposedly caused by strong wind, which removes the small, fresh fallen snow grains and exposes deeper layers of

10    larger grains. With mean wind speeds of $4\,\mathrm{m\,s^{-1}}$ and maximum wind gusts reaching $11\,\mathrm{m\,s^{-1}}$ at Kohnen station, drifting snow occured mainly due to creeping or saltation of the ice crystals. This wind-induced transportation of fresh fallen snow grains is superimposed on the signal of snow metamorphism in the temporal evolution of SSA retrieved from CORAS.

    The mean SSA is $42\pm13\,\mathrm{m^2\,kg^{-1}}$. The SSA retrieved from CORAS measurements reproduces the in situ probing (solid black line in Fig. 6) within the measurement uncertainties and range within the shaded gray area, which indicates the standard

deviation of the mean SSA value along the 100 m-transect where SSA was probed. The standard deviation is a measure of the small-scale variability in SSA mainly caused by wind-induced roughness structures of the snow surface. Nevertheless, the temporal signal in SSA is significant for all sample positions. Only during the end of the campaign, the last two snowfall-induced maxima in SSA are overestimated by CORAS. The agreement between the SSA retrieved by CORAS and the in situ

data is reflected in the linear correlation coefficient of 0.81 (95 % confidence interval: 0.66-0.89).

On eight days throughout the campaign, no retrieval of SSA using CORAS data was possible due to broken clouds. For that reason, the first maximum of the in situ measured SSA (15 December 2013) could not be reproduced by CORAS data. Overcast retrieval conditions were used on four days (filled circles: 3, 6, 10, and 12 January 2014). The retrieved SSA on overcast days agree well with the in situ measurements and are in coherence with the retrieved SSA under cloudless conditions.

This illustrates the potential of extending the retrieval method by applying the spherical albedo. In addition, it highlights the benefit of ground-based observations that, in comparison to satellite observations, provide continuous time series and are not restricted to cloudless time periods only.

## 4.2   SSA and optical snow grain size from SMART (airborne)

On five days between 27 December 2013 and 2 January 2014, SSA and $r_{\mathrm{opt}}$ were retrieved from airborne spectral albedo mea-

surements by SMART (green circles in Fig. 6). With a mean value of $29\pm1\,\mathrm{m^2\,kg^{-1}}$, SMART seems to slightly underestimate in situ SSA by a factor of 1.2. Using the same calibration reference and the identical retrieval algorithm for both CORAS and SMART, the differences are likely connected to the different sizes of the sampling areas covered by both instruments. While CORAS measurements represent a spot of about $2\,\mathrm{x}\,2\,\mathrm{m^2}$, SMART measurements have a larger footprint and were averaged over an area of $5\,\mathrm{x}\,5\,\mathrm{km^2}$ surrounding Kohnen station. Furthermore, the area over which the airborne data is averaged

is largely determined by the flight track. On such scales, the local small-scale variability of SSA as indicated by the in situ measurements can lead to significant differences in SSA. Already along the 100 m-transect, in situ SSA varied by up to 34 % (4 January 2014). On larger scales, this variability is likely to increase. In addition, unidentified systematical errors in the airborne SMART measurements are a possible reason for the underestimation of SSA and deserve further attention in future studies.

## 4.3   SSA and optical snow grain size from MODIS (spaceborne)

A smaller bias is present in the SSA retrieved from MODIS data ($37\pm11\,\mathrm{m^2\,kg^{-1}}$, red line in Fig. 6), which integrates over an area of $50\,\mathrm{x}\,50\,\mathrm{km^2}$ surrounding Kohnen station. The uncertainty in Fig. 6 is given as the standard deviation of SSA over the $50\,\mathrm{x}\,50\,\mathrm{km^2}$. The SSA derived from MODIS observations could reproduce the SSA signal from in situ measurements, the linear correlation coefficient is with 0.86 (95% confidence interval: 0.75-0.92) slightly higher than for the CORAS measurements. Furthermore, it was able to resolve abrupt changes in SSA due to precipitation or wind-induced transportation of snow grains.

For high solar zenith angles, the SGSP algorithm is known to underestimate the actual optical snow grain size (Zege et al., 2011). During the entire observation period, the solar zenith angle varied between $52°$ and $87°$ and in 48 % of the time was higher than $70°$. However, the optical snow grain size retrieved from MODIS data mostly showed a slight overestimation compared to the in situ measurements. The comparison of the ground-based (CORAS) and spaceborne (MODIS) remote

sensing methods to retrieve SSA yields a linear correlation coefficient of 0.77 (95 % confidence interval: 0.61-0.87) which lies in the same range as the correlation coefficient between CORAS and the in situ measurements.

## 5  Conclusions

The temporal variability of SSA and respective $r_{opt}$ on the East Antarctic plateau were investigated during austral summer 2013/14 utilizing spectral albedo measurements (ground-based and airborne) and MODIS reflectance measurements. The retrieved SSA and $r_{opt}$ were compared with in situ data.

For the retrievals from spectral surface albedo measurements, the SGSP algorithm was extended to spectral ratios of albedo at 1280 nm and 1100 nm wavelength. Being independent of systematic measurement uncertainties (e.g., cross-calibration of the optical inlets), this approach decreases the uncertainty of the retrieved SSA compared to the single-wavelength approach. The retrieval was successfully applied to measurements in overcast conditions by using the spherical instead of the plane albedo. Satellite observations are limited by clouds in space and time. During the two months of observations at Kohnen station, cloudless conditions were present only 264 h out of 1272 h total observation time (21 %). However, many cloudy periods were characterized by a broken cloud field (62 % of total observation time). In this case, the concept of spherical albedo is not applicable and SSA retrievals might fail. Therefore, only overcast conditions were included in the analysis.

SSA retrieved from CORAS measurements varied between $27 \, m^2 \, kg^{-1}$ and $89 \, m^2 \, kg^{-1}$ and revealed distinct maxima related to light snowfall at Kohnen station. The average increase in SSA due to snowfall was $28 \, m^2 \, kg^{-1}$. The maxima were followed by a gradual decrease in SSA, which was partly caused by snow metamorphism and by wind-induced transport of the fresh fallen ice crystals. During the longest dry period (18 until 30 December 2013), SSA decreased on average by $3.2 \, m^2 \, kg^{-1}$ per day. This corresponded to a daily increase of $r_{opt}$ by 5.8 µm.

The temporal evolution of SSA retrieved from the ground-based CORAS measurements could reproduce the in situ measurements (linear correlation coefficient: 0.81). The same holds true for the spaceborne MODIS retrieval (0.86). Despite the biases in SSA retrieved from the different instruments, the agreement especially between SSA retrieved from CORAS and in situ measurements emphasizes the potential of the retrieval algorithms. SSA retrieved from airborne SMART measurements underestimated in situ SSA by a factor of 1.2. This might be due to spatial averaging. However, especially the differences between SSA derived from CORAS and SMART measurements need to be investigated in more detail in further research, before extending the retrieval validation between SMART and MODIS to a larger area in Dronning Maud Land which would also cover coastal areas with supposedly higher variability in SSA. An Antarctic-wide survey of albedo mapping can only be achieved by satellite observations. For this purpose, a more detailed understanding of the relation between satellite and in situ observation of SSA is required, including the influence of the different spatial scales of satellite, airborne, and in situ measurements.

The validation presented in this study provided an unique testbed for retrievals of optical snow grain size from satellite reflectance and spectral surface albedo measurements under Antarctic conditions where in situ data are scarce and can be used for testing prognostic snowpack models in Antarctic conditions.

*Data availability:* We published the three time series of SSA retrieved from ground-based CORAS, airborne SMART, and spaceborne MODIS measurements as shown in Fig. 6 in the Publishing Network for Geoscientific & Environmental Data (PANGAEA). The data set is available under: https://doi.pangaea.de/10.1594/PANGAEA.880815 (Carlsen et al., 2017).

*Acknowledgements.* This work was supported by the Deutsche Forschungsgemeinschaft (DFG) in the framework of the priority programme "Antarctic Research with comparative investigations in Arctic ice areas" (SPP 1158) by the grants WE1900/29-1 and BI 816/4-1. We gratefully acknowledge the support by the SFB/TR 172 "ArctiC Amplification: Climate Relevant Atmospheric and SurfaCe Processes, and Feedback Mechanisms $(\text{AC})^3$" funded by the DFG. We are grateful to the Alfred Wegener Institute, Helmholtz Centre for Polar and Marine Research, Bremerhaven, Germany for supporting the campaign with logistics, the aircraft and manpower in Antarctica. In addition, we would like to thank Kenn Borek Air Ltd., Calgary, Canada for the great pilots who made the complicated measurements possible.

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

**Table 1.** List of instruments operated on ground and on Polar 6 aircraft.

|  | Instrument | Measured quantity | Specifications |
|---|---|---|---|
| **Ground-based** (Kohnen) | Kipp & Zonen CM22 | $F^{\downarrow}$, $F^{\uparrow}$ ($\mathrm{W\,m^{-2}}$) | Broadband, 0.2-3.6 µm |
|  | Eppley Precision Infrared Radiometer | $F^{\downarrow}$, $F^{\uparrow}$ ($\mathrm{W\,m^{-2}}$) | Broadband, 3.5-50 µm |
|  | CORAS | $F^{\downarrow}(\lambda)$, $F^{\uparrow}(\lambda)$ ($\mathrm{W\,m^{-2}\,nm^{-1}}$) | Spectral, 0.3-2.2 µm |
|  | CANON EOS 6D | $I^{\uparrow}$ ($\mathrm{W\,m^{-2}\,nm^{-1}\,sr^{-1}}$) | 180° fish-eye lens |
|  | CANON EOS 600D | All-sky images |  |
|  | CANON EOS 600D (2x) | Photogrammetric images | Image overlap: 50 % |
|  | Sun photometer SP1A31 | AOD | 10 channels: 368.5 to 1019.4 nm |
|  | Automatic weather station | $p$, $T$, RH, $F^{\uparrow}$, $F^{\downarrow}$, $\vec{v}$, snow accumulation | 1 min averages |
|  | Ultrasonic anemometer | $\vec{v}$ |  |
|  | Radio sounding | $p$, $T$, RH, $\vec{v}$ |  |
|  | Synoptic observations | Cloud cover, precipitation | Visual observation |
|  | IceCube by A2 Photonic Sensors | SSA ($\mathrm{m^2\,kg^{-1}}$), $\rho_{\mathrm{snow}}$ ($\mathrm{kg\,m^{-3}}$) | Wavelength: 1310 nm |
|  | Pt-100 | Snow temperature profile |  |
| **Airborne** (Polar 6) | Kipp & Zonen CMP22 | $F^{\downarrow}$, $F^{\uparrow}$ ($\mathrm{W\,m^{-2}}$) | Broadband, 0.2-3.6 µm |
|  | Kipp & Zonen CGR4 | $F^{\downarrow}$, $F^{\uparrow}$ ($\mathrm{W\,m^{-2}}$) | Broadband, 4.5-42 µm |
|  | SMART | $F^{\downarrow}(\lambda)$, $F^{\uparrow}(\lambda)$ ($\mathrm{W\,m^{-2}\,nm^{-1}}$) | Spectral, 0.3-2.2 µm |
|  |  | $I^{\downarrow}(\lambda)$, $I^{\uparrow}(\lambda)$ ($\mathrm{W\,m^{-2}\,nm^{-1}\,sr^{-1}}$) | Spectral, 0.3-2.2 µm |
|  | CANON EOS 1D Mark III | $I^{\uparrow}$ ($\mathrm{W\,m^{-2}\,nm^{-1}\,sr^{-1}}$) | 180° fish-eye lens |
|  | RIEGL VQ580 | Surface topography | Airborne laser scanner |
|  | AIMMS20 | $\vec{v}$, $p$ | Meteorological measurements |
|  | KT19 | $T_{\mathrm{surf}}$ | Radiation thermometer |

$F^{\downarrow}$: downward irradiance, $F^{\uparrow}$: upward irradiance, $\lambda$: wavelength (indicates spectral quantity), $I^{\uparrow}$: reflected radiance, AOD: aerosol optical depth, $p$: pressure, $T$: temperature, RH: relative humidity, $\vec{v}$: wind vector, SSA: specific surface area, $\rho_{\mathrm{snow}}$: snow density, $T_{\mathrm{surf}}$: surface temperature