# Peer review of "Comparison of different methods to retrieve optical-equivalent snow grain size in central Antarctica"

_The Cryosphere, 2016_

## Referee Comment (RC1) · E. Zege (Referee) · 2 Mar 2017

**Review of the manuscript**

**"Comparison of different methods to retrieve  effective snow grain size in central Antarctica"**

**by T. Carlsen, G. Birnbaum, A. Ehrlich, J. Freitag, G. Heygster3, L.Istomina,S. Kipfstuhl, A. Orsi, M. Schäfer, and M. Wendisch**

**General comments**

The problem discussed in the manuscript is very topical and important for a few modern challenges, including the remote sensing of polar regions, the climate change problem, etc. Joint consideration of the satellite, airborne and field techniques to explore snow microstructure in Central Antarctica is presented in this manuscript. This task required the joint efforts of scientists from several European research centers and has brought valuable results.  The ground-based experimental procedures included measurements of the SSA (specific surface area) of snow (technique of Gallet at al,   The Cryosphere, 3, 2009) and of the snow spectral albedo at wavelengths 1280nm and 1100nm. The 2-wavelength technique for retrieving effective snow grain size with this data has been refined. As a result  the admissible accuracy of the used retrieval procedure has been secured.
For validation of the currently employed retrieval techniques the results from MODIS and airborne SMART system (AWI) taken simultaneously to the field measurements were explored as remote sensing data. The results of the validation of the various retrieval techniques presented in this study look very promising. For instance, very satisfactory correlation between field data (SSA) and results of the SGSP retrieval from the spaceborne MODIS data (linear correlation coefficient: 0.86) has been found.
 The article is written clearly, well-structured, presents new useful results, and demonstrates authors' good knowledge of the state of the art of the problem.
The problems considered in this manuscript are completely within the scope of TC
Below a list of specific comments is presented.

 **This paper is recommended to be published in "The Cryosphere" after minor revision.**

**Specific comments**

Below the pieces of text from the reviewed manuscript will be given in italic in difference on the reviewer comments.

Page 8, Lines 3-6:

*The large uncertainty of the SGSP retrieval for high solar zenith angles is related to the conversion of the measured reflectance from one viewing direction by the satellite sensor into measured albedo. For this, the bidirectional reflectance distribution function (BRDF) of the snow surface has to be assumed."*

This statement is incorrect*. "The conversion of the measured reflectance from one viewing direction by the satellite sensor into measured albedo "* **is not used in the SGSP algorithm.** Actually in the SGSP algorithm, the angular dependency in the registered signals is excluded using registration of

the signal for additional wavelength, and no assumptions about BRDF is used.  It is one of the main advantages of this technique.

P.8, Lines 6- 7
*Due to the strong forward scattering characteristic for snow grains, **small errors in the assumed BRDF greatly distort the albedo**, especially at low sun elevations*

Correspondingly, this sentence should be corrected, because there is no use of "assumed BRDF "in SGSP procedure.

P.10, Fig 5

Fig. 5 presents important data. For more fast understanding and analysis it would be very useful to give the corresponding Sun polar angles as the second scale at X- axis.

P.10, Lines 7-8

*As no snowfall occurred on that day, the diurnal cycle is likely to be an  artifact  originating from the change in solar zenith angle and the assumed escape function $K_0(\theta_0)$.*

It might be a very interesting and useful observation.  In all cases it makes a reader think about necessity to use more accurate approximation of the $K_0(\theta_0)$ function for interpretation data at low Sun positions.  The recommended   formula

$$K_{emp}(\theta_0) = \frac{3}{7}(1.5 + 1.1\cos\theta_0) = \frac{2}{7}(1 + 0.73\cos\theta_0) \quad (10)$$

should be rather taken as empirical approximation for considered case.  This result requires a more detail consideration in future studies.

P.10, Lines 8-10

*The escape function might be incorrect if the snow BRDF is more complex due to the non-spherical snow grain shape.*

This statement is incorrect.  More accurate would be "The used approximation   (3) for escape function might be incorrect for snow at very oblique incidence because of its very elongated  phase   function".

P.11, Lines 24-25

*Therefore, each instrument retrieves the effective grain size from a different depth within the snowpack.*

Moreover, even the same instrument takes measurements from layers with different depth dependent on wavelength.  All retrievals considered in this paper  (and in  many others) were performed neglecting snow layer stratification.

 P.13, Line 4

*which can be related to snowfall of about **1mm** at Kohnen*

Do authors really mean "1mm"?

**Minor correction**

Abstract

P.1

*The effective size of snow grains affects the reflectivity of snow surfaces and thus the local surface energy budget in particular in polar regions. Therefore, the specific surface area (SSA) was monitored.*

There is some breach of logic: " *The effective size of snow grains affects… Therefore, the specific surface area (SSA) was monitored.* " Still it is not stated how these quantities are related.

P.2, Line 5

*For example, Munneke et al. (2008) found variations of **the broadband albedo** of snow at five different locations in Dronning Maud Land in a range between 0.77 and 0.88.*

I recommend small change:

" For example, Munneke et al. (2008) found variations of the broadband albedo of snow in a range between 0.77 and 0.88 at five different locations in Dronning Maud Land."

P.2, Line 3-7

*However, the snow surface albedo varies both on a temporal and spatial scale. For example, Munneke et al. (2008) found variations of **the broadband albedo** of snow at five different locations in Dronning Maud Land in a range between 0.77 and 0.88.*

*This variability is caused by different parameters such as snow grain size (and shape), surface roughness (e.g., Warren et al., 1998), soot content (e.g., Bond et al., 2013), and cloudiness; **it depends on wavelength…**and solar position.*

Because the first sentence is only about the broadband albedo, the second sentence is required to be corrected.

---

## Referee Comment (RC2) · G. Picard (Referee) · 19 May 2017

General comments

The paper compares four time-series of surface snow grain size (specific surface area to be more precise) obtained around Kohnen station in Antarctica derived from different albedo measurements: one is from in situ manual, one from in situ automatic, another one using an airborne sensor, and the last one using MODIS satellites. The paper describes the datasets, the methodology to retrieve the grain size from the different sensors, then the results and provide a short discussion and a conclusion.

The paper theme is of great interest as the grain size controls the albedo which is a significant driver of the surface energy budget in Antarctica. Because time-series of grain size are scarce, in Antarctica and elsewhere, comparison of methodologies to

obtain them is an important ongoing effort in the snow community.

The paper is well written and easy to follow except in a few places where critical information are missing (see detailed comments). This requires minor work but is necessary for the read to understand the assumptions used by the authors, and to allow reproducibility and comparison with other studies.

Overall the paper is quite short. The authors should address in their response to reviewers the reason why this study is not merged with the paper in preparation (Freitag et al.). The review of the literature is also relatively light and should be completed (see some personal suggestions below, but works from other groups should be considered as well). The discussion should be completed with an analysis of the results with respect to other previous studies.

Uncertainties are given at different stages of the methodology, which is very useful, but the way these uncertainties have been estimated is not described, and they seems to me often overoptimistic. Even though it is notoriously difficult to estimate uncertainties, some justification are required. In addition, the authors do not consider the effect of the roughness, and they neglect the impact of wavelength-dependent errors of spectra calibration on the ratio R. Both are probably not negligible and should be evaluated or at least indicated.

The paper does not show albedo spectra from which SSA are derived. Given the focus of the paper, it could be acceptable. Nevertheless I suggest to add example of spectra for one date to illustrate the derivation of SSA from albedo and to possibly help in the analysis of the bias observed between the airborne timeseries and other time-series which remains unexplained.

To foster comparison in the future, I recommend to publish the four timeseries once the paper is accepted.

Detailed comments:

Abstract:

- The term "effective size of snow grain" is not well defined, especially "effective" is relative to the domain (optics, microwave, chemistry, etc). Optical grain size is more adequate.

- The abstract tells what has been done, but is not an abstract of the paper. The objective of the study is missing, as well as the conclusion.

Introduction

P2 L3 "most of the sea ice are covered with snow with little seasonal variability". The variability of sea ice is huge in Antarctica.

P2 L9 "Furthermore, snow surface albedo varies with snow depth and the liquid water content.". Need a reference for each effect. The dependence to snow depth is almost irrelevant in Antarctica.

P2 L11: Add references of published works (Libois et al., Munneke et al., Picard et al., Pirrazini et al., Warren and co, Zender and co, etc).

P2 L18: Are these values here for optical grain size ? Is the reference about optical grain size or other grain size metrics ? The sentence should be more precise about this.

P 3 L7: It is not clear if the sentence is about the impact of the albedo-grain shape or BRDF-grain shape. Both are quite different and have different impact depending on the algorithm used for the retrieval. For the former issue, the value of 2.6 is extreme. See Kokhanovsky and Zege 2004, Picard et al. 2009, Libois et al. 2013 for other works on this question.

About P3, Libois et al. 2015 and Picard et al. 2016 (both in The Cryosphere), have produced time-series of SSA (3 years) at Dome C in Antarctica which seems relevant to the present study. See also Jin et al. 2008 and Picard et al. 2012 for earlier (and

shorter) time-series of SSA in-situ and satellites measurements.

P3 L12-13: "However, polar orbiting satellites do not provide a sufficiently high temporal resolution that may resolve snow precipitation events and snow metamorphism that typically can advance in a matter of hours.". With the number of satellites (MODIS, Sentinel, SSM/I, AMSU, ...) and the convergence of the orbits in the polar regions, sub daily resolution can be achieved.

P4L8: According to Gallet et al. 2009, IceCUBE (i.e. DIFUSSS with 1300 nm only) can not be used from SSA of 60m2/kg and above. If the "in prep" paper shows different results, it would be interesting to add a sentence here. Otherwise the limitation should be indicated.

P4 L15: invert the temperatures

Fig 1: I'm not convinced that this first figure (the octa plot) is useful for the paper. It shows raw data that are not used as is in the following and the interpretation is short and inconclusive. Instead, I suggest to plot a time-series (a normal one as for the air temperature, not a time versus date graph) cloudy/not cloudy as used by the algorithm to switch between diffuse/direct radiation.

P5L1: "polar day". This term is ambiguous and the sentence is not strictly logical, air temperature decrease observed before the end of the "polar day" is due to a change of solar elevation, not a change of day duration.

P5L10: where these values of uncertainty are coming from ? As explained in the general comment, the description of the CORAS instrument and the evaluation of the uncertainty are missing. This part needs to be expanded but at least one paragraph. Please also consider the shadowing effect, frost formation, ... Information such as the cosine correction method, the frequency of observation, and other instrumental details are needed as well.

P7L20: Regarding the choice of the values of B and g in the "middle of the range":

1) Values obtained from measurements are now available. Libois et al. 2013 and 2014 have measured B and recommended some values of g based on measurements.

2) The uncertainty of these values impacts the SSA estimation and needs to be taken into account in the evaluation of the uncertainties proposed in this paper which only considers instrumental uncertainties.

P8L5-L8: A few critical information are missing about the MODIS algorithm. - Which BRDF parameterization has been used and how ? - How the radiances have been converted to ground-level surface reflectances ? - Has an atmospheric model been used ? - Which product and version have been used here ?

P9L13: The argument implicitly assumes that the calibration error is wavelength-independent, which is often not the case. Even if the ratio R is less sensitive to calibration issues, the residual wavelength dependence can greatly affects the estimation of SSA as shown in Picard et al. 2016 (e.g. cosine correction, 100% direct assumption...).

P8L16: these values have been obtained for perfect clear-sky. To my personal experience, direct/diffuse ratio shows huge variations in the infrared due to thin cloud (more than in the visible because the diffuse component is already quite large). Calculations of the atmosphere model with cloud with varying optical depth and crystal size would be useful, because the estimation of SSA using ratio (i.e. Eq 9) is sensitive to the spurious wavelength-dependence resulting from cloud cover.

P10L7: I suggest to improve a little the justification of the constance of SSA during the day. E.g. doi:10.5194/tc-8-1205-2014

P10L10: the surface roughness/slope at the footprint scale is another likely player that affects the theory used here. See e.g. Dumont et al. 2017 in TC (see the slope estimation).

P14L5: the argument on the scales seems weak because the airborne data are between the in-situ and MODIS data in terms of scales and MODIS shows little bias w/r

to in-situ measurements. I guess that the authors have analyzed in details the spectra and various sources of errors from the different sensors without success. May be add part of this analysis. For instance, showing coincident spectra from the different sensors could help to understand the differences, and at least to rule out some potential defects of the sensors.

---

## Author Comment (AC1) · 31 Jul 2017

**Reply to Referee reviews**

**Manuscript title**

Comparison of different methods to retrieve effective snow grain size in central Antarctica

**Date of reply**

31 July 2017

Dear Reviewers,

thank you for the helpful comments and suggestions on improving the manuscript.

The detailed replies to your comments (highlighted in gray) are given below, followed by the specific changes made in the manuscript (changes are highlighted in bold text). The marked-up version (see below) illustrates all changes made compared

10 to the manuscript published in The Cryosphere Discussions.

Kind regards from the authors.

**Replies to comments of Referee 1 (E. Zege)**

Page 8, Lines 3-6:

*The large uncertainty of the SGSP retrieval for high solar zenith angles is related to the conversion of the measured reflectance from one viewing direction by the satellite sensor into measured albedo. For this, the bidirectional reflectance distribution function (BRDF) of the snow surface has to be assumed.*

This statement is incorrect. 'The conversion of the measured reflectance from one viewing direction by the satellite sensor into measured albedo' is not used in the SGSP algorithm. Actually in the SGSP algorithm, the angular dependency in the registered signals is excluded using registration of the signal for additional wavelength, and no assumptions about BRDF is used. It is one of the main advantages of this technique.

Thank you for pointing out this formulation mishap. What we intended to say here is related to the surface albedo model and

15 not the conversion of the measured radiance into albedo. The assumption on the particle form factor and approximation of the photon escape function which defines the angular behavior of the light reflected from a medium can be less accurate for oblique illumination angles. Similarly, as can be seen from Zege et al. (2011) Fig. 1, the slight dependence of the phase function on the wavelength is angle dependent. This means that for some angles the assumption of the spectrally constant BRDF shape holds

better than for others, which may explain the diurnal cycle of the retrieved snow grain size discussed below. These details have been included in the manuscript.

**Changes in manuscript:**

5   After a preliminary separation of snow pixels, the effective snow grain size of each pixel is retrieved from radiance measurements of MODIS channels 3 (469 nm wavelength), 2 (858 nm), and 5 (1240 nm). **Assuming a spectrally constant bidirectional reflectance distribution function (BRDF) of snow and using the combination of three spectral channels, the angular dependency of the measured radiance is excluded.**

(...)

10   **The large uncertainty of the SGSP retrieval for high solar zenith angles is related to the assumptions on the particle form factor $A$ and the approximation of the escape function $K_0(\theta_0)$ which is less accurate for oblique illumination angles. In combination with the strong forward scattering characteristic for snow grains, small errors in the assumed $A$ and $K_0(\theta_0)$ can greatly distort the albedo. Similarly, the spectral behavior of the BRDF of snow slightly depends on the illumination angle (Zege et al., 2011).** Consequently, within this work the satellite retrieval is limited to $\theta_0 \leq 75°$.

P.8, Lines 6- 7:

*Due to the strong forward scattering characteristic for snow grains, small errors in the assumed BRDF greatly distort the albedo, especially at low sun elevations*

Correspondingly, this sentence should be corrected, because there is no use of 'assumed BRDF' in SGSP procedure.

15   Similar to above, this sentence referred to the assumptions on particle form factor and the assumption that the BRDF shape does not change with the wavelength. This has been reformulated (see above).

P.10, Fig 5:

Fig. 5 presents important data. For more fast understanding and analysis it would be very useful to give the corresponding Sun polar angles as the second scale at X- axis.

Thank you for this suggestion. We adjusted Fig. 5 accordingly.

**Changes in manuscript:**

[Figure]

Figure 5 shows the results retrieved from ground-based measurements with CORAS (SSA and effective snow grain size) on 24 December 2013 between 6 UTC and 18 UTC. **During this time, the solar zenith angle varied between 52° and 67 °.**

P.10, Lines 7-8:

*As no snowfall occurred on that day, the diurnal cycle is likely to be an artifact originating from the change in solar zenith angle and the assumed escape function $K_0(\theta_0)$.*

It might be a very interesting and useful observation. In all cases it makes a reader think about necessity to use more accurate approximation of the $K_0(\theta_0)$ function for interpretation data at low Sun positions. The recommended formula

$$K_{\mathrm{emp}}(\theta_0) = \frac{3}{7}(1.5 + 1.1\cos\theta_0) = \frac{2}{7}(1 + 0.73\cos\theta_0)$$

should be rather taken as empirical approximation for considered case. This result requires a more detail consideration in future studies.

Yes, we agree and have corrected the manuscript accordingly. The equation given in the manuscript was derived purely from empirical analysis of the diurnal time series of this single case and is likely not valid in other cases. For a more general adjustment of the escape function, a more careful consideration must indeed be performed. However, this is out of the scope of this work, but should be addressed in future studies using explicit BRDF measurements.

**Changes in manuscript:**

**The analysis of measurements during** other cloudless days showed similar features. Therefore, $K_{\mathrm{emp}}$ was applied for the entire period of measurements. **However, it should be noted that the escape function $K_{\mathrm{emp}}$ from Eq. 13 is an empirical approximation for the cases investigated here. To derive a more general description of the escape function, more cases and explicit BRDF measurements are needed, which is beyond the scope of this study.**

P.10, Lines 8-10:

*The escape function might be incorrect if the snow BRDF is more complex due to the non-spherical snow grain shape.*

This statement is incorrect. More accurate would be 'The used approximation (3) for escape function might be incorrect for snow at very oblique incidence because of its very elongated phase function'.

Thank you for this comment. Indeed, the oblique angles are problematic because of the strong forward scattering phase function and any inaccuracies in the approximation will have a greater effect at these angles. At the same time, we meant to highlight the importance of the correct selection of the form factor, that is, the asymmetry parameter and the absorption enhancement parameter at this point. We have changed the text to accommodate for these details.

**Changes in manuscript:**

**The approximation for the escape function (Eq. 4) might be incorrect for snow at oblique incidence due to its elongated phase function. At the same time, a representative form factor $A$ is required to account for the non-spherical snow grain shape.**

P.11, Lines 24-25:

*Therefore, each instrument retrieves the effective grain size from a different depth within the snowpack.*

Moreover, even the same instrument takes measurements from layers with different depth dependent on wavelength. All retrievals considered in this paper (and in many others) were performed neglecting snow layer stratification.

This statement is true. We included this comment to foster reading comprehension.

**Changes in manuscript:**

**All retrievals considered in this study were performed neglecting snow layer stratification.**

P.13, Line 4:

*which can be related to snowfall of about **1mm** at Kohnen*

Do authors really mean '1mm'?

Yes, we indeed mean snowfall of about $1\,\mathrm{mm}$. This is an approximate value which stems from visual estimation as no quantitative measurements were performed during the observations. In the dry atmosphere over the high Antarctic plateau, precipitation events with less than a few $\mathrm{mm}$ depth are common.

**Changes in manuscript:**

(...) which can be related to snowfall of about $1\,\mathrm{mm}$ at Kohnen **(visual estimation)**.

Abstract P.1:

*The effective size of snow grains affects the reflectivity of snow surfaces and thus the local surface energy budget in particular in polar regions. Therefore, the specific surface area (SSA) was monitored.*

There is some breach of logic: 'The effective size of snow grains affects... Therefore, the specific surface area (SSA) was monitored.' Still it is not stated how these quantities are related.

Thank you for pointing this out, we changed this part of the abstract to avoid this breach of logic.

**Changes in manuscript:**

**The size of snow grains** affects the reflectivity of snow surfaces and, thus, the local surface energy budget in particular in polar regions. Therefore, the specific surface area (SSA) **as an equivalent measure for snow grain size** was monitored for a

5    two-month period in central Antarctica (Kohnen research station) during austral summer 2013/14.

P.2, Line 5:

*For example, Munneke et al. (2008) found variations of the broadband albedo of snow at five different locations in Dronning Maud Land in a range between 0.77 and 0.88.*

I recommend small change:

'For example, Munneke et al. (2008) found variations of the broadband albedo of snow in a range between 0.77 and 0.88 at five different locations in Dronning Maud Land.'

Thank you. We changed it as suggested.

P.2, Line 3-7:

*However, the snow surface albedo varies both on a temporal and spatial scale. For example, Munneke et al. (2008) found variations of **the broadband albedo** of snow at five different locations in Dronning Maud Land in a range between 0.77 and 0.88.*

*This variability is caused by different parameters such as snow grain size (and shape), surface roughness (e.g., Warren et al., 1998), soot content (e.g., Bond et al., 2013), and cloudiness; **it depends on wavelength**... and solar position.*

Because the first sentence is only about the broadband albedo, the second sentence is required to be corrected.

This is correct, thank you for pointing it out. We adjusted the manuscript accordingly.

**Changes in manuscript:**

**Changes of broadband and spectral snow albedo are** caused by different parameters such as snow grain size (and shape),

10    surface roughness (e.g., Warren et al., 1998), soot content (e.g., Bond et al., 2013), and cloudiness. **It further depends** on wavelength (e.g., Hudson et al., 2006; Warren and Brandt, 2008) **as well as** solar position (e.g., Wiscombe and Warren, 1980;

Wiscombe, 1980; Dumont et al., 2010) **and varies with snow depth (e.g., Wiscombe and Warren, 1980) and liquid water content (e.g., Wiscombe and Warren, 1980; Gallet et al., 2014)**.

**Replies to comments of Referee 2 (G. Picard)**

Overall the paper is quite short. The authors should address in their response to reviewers the reason why this study is not merged with the paper in preparation (Freitag et al.). The review of the literature is also relatively light and should be completed (see some personal suggestions below, but works from other groups should be considered as well). The discussion should be completed with an analysis of the results with respect to other previous studies.

The reason why these two studies are not merged are the different focuses of both studies. While the manuscript discussed here compares methods to retrieve effective snow grain size from different remote sensing platforms, the manuscript in preparation by Freitag et al. focuses on the seasonal evolution of SSA at Kohnen station and concentrates on the microphysical processes within the snowpack. Merging both studies would produce synergy effects but rather weaken the visibility of each individual study. In our manuscript, the in situ dataset is only used for validation of the remote sensing results. Of course, time series and changes of SSA are discussed but any analysis of the microphysical processes causing these changes are beyond the scope of our study. In the opinion of the authors, this justifies separate manuscripts.

The comments on the literature review and the discussion are considered within the specific comments.

Uncertainties are given at different stages of the methodology, which is very useful, but the way these uncertainties have been estimated is not described, and they seems to me often overoptimistic. Even though it is notoriously difficult to estimate uncertainties, some justification are required. In addition, the authors do not consider the effect of the roughness, and they neglect the impact of wavelength-dependent errors of spectra calibration on the ratio R. Both are probably not negligible and should be evaluated or at least indicated.

We have revised the uncertainty estimation and its description to make clear where the uncertainties of each measurement result from. The changes made in the manuscript are stated at the corresponding specific comments.

The paper does not show albedo spectra from which SSA are derived. Given the focus of the paper, it could be acceptable. Nevertheless I suggest to add example of spectra for one date to illustrate the derivation of SSA from albedo and to possibly help in the analysis of the bias observed between the airborne timeseries and other time-series which remains unexplained.

We agree that showing a measured spectral albedo can further illustrate the derivation of SSA. We therefore included a spectral albedo measured by CORAS on 24 December 2013 (14 UTC) in Fig. 4a and adapted the simulated snow surface plane albedos to the solar zenith angle of the time of measurement (around 54°). In addition, we modified Fig. 4b as follows: for the illustration of retrieved $r_{\text{eff}}$ from the ratio $\mathcal{R}$ we now use the ratio computed from the spectral albedo by CORAS as shown in Fig. 4a.

**Changes in manuscript:**

[Figure]

Left (a): Surface plane albedo for effective snow grain sizes between $10\,\mu m$ and $200\,\mu m$, $\theta_0 = 54°$, $A = 5.8$, $\chi$-data from Warren and Brandt (2008). M3, M2, and M5 mark MODIS spectral bands used within the SGSP algorithm. $\lambda_1$ and $\lambda_2$ denote wavelengths used within the CORAS and SMART grain size retrieval. **The read solid line and shaded gray show a spectral albedo measured by CORAS on 24 December 2013 (14 UTC).** Right (b): Illustration of retrieval principle. dependence of ratio $\mathcal{R}$ with respect to effective snow grain size for different solar zenith angles (50° to 80°) and for overcast conditions, $A = 5.8$, $\chi$-data from Warren and Brandt (2008). Blue and red lines illustrate the retrieval of effective snow grain size from **a measured albedo ratio $\mathcal{R} = 0.7$ with a relative uncertainty of 5.5 %**.

**Furthermore, a spectral albedo between $700\,nm$ to $1600\,nm$ wavelength measured by CORAS on 24 December 2013 (14 UTC) is shown. The data gap between $1300\,nm$ and $1400\,nm$ wavelength is resulting from low signal-to-noise ratios at these wavelengths.**

Applying Eq. 12 to the **spectral albedo measured by CORAS on 24 December 2013 (Fig. 4a), the measured ratio $\mathcal{R}$ of $0.702\pm0.039$ leads to an estimated effective snow grain size of about $90\pm31\,\mu m$ at $\theta_0 = 54°$ (blue and red lines in Fig. 4b).**

To foster comparison in the future, I recommend to publish the four timeseries once the paper is accepted.

This is a valuable comment, thank you. Indeed, we have already planned to publish the following three time series of SSA in the Publishing Network for Geoscientific & Environmental Data (PANGAEA) : CORAS, SMART, MODIS.

This will be done as soon as the manuscript is published.

Abstract:

The term 'effective size of snow grain' is not well defined, especially 'effective' is relative to the domain (optics, microwave, chemistry, etc). Optical grain size is more adequate.

This is correct. The definition of the term 'effective size of snow grains' was described too late. We therefore refrain from using 'effective' within the abstract already. Instead, we introduce the study in the abstract on the basis of the SSA alone.

Additionally, we carefully revised the use of 'effective' . If the quantity of 'effective snow grain size' is used, 'effective' is included. If, in general, we refer to the size of snow grains without explicitly pointing to a physical quantity, 'effective' is omitted.

**Changes in manuscript:**

5  **The size of snow grains** affects the reflectivity of snow surfaces and, thus, the local surface energy budget in particular in polar regions. Therefore, the specific surface area (SSA) **as an equivalent measure for snow grain size** was monitored for a two-month period in central Antarctica (Kohnen research station) during austral summer 2013/14.

(...)

10  (...)

Abstract:

The abstract tells what has been done, but is not an abstract of the paper. The objective of the study is missing, as well as the conclusion.

We revised the abstract accordingly.

**Changes in manuscript:**

The size of snow grains affects the reflectivity of snow surfaces and, thus, the local surface energy budget in particular in polar
15  regions. Therefore, the specific surface area (SSA) **as an equivalent measure** for snow grain size was observed for a two-month period in central Antarctica (Kohnen research station) during austral summer 2013/14. The data were retrieved on the basis of ground-based spectral surface albedo measurements collected by the COmpact RAdiation measurement System (CORAS) and airborne observations with the Spectral Modular Airborne Radiation measurement sysTem (SMART). **The Snow Grain Size and Pollution amount (SGSP) algorithm, originally developed to analyze spaceborne reflectance measurements by**
20  **the MODerate Resolution Imaging Spectroradiometer (MODIS), was modified in order to reduce the impact of the solar zenith angle on the retrieval results and to cover measurements in overcast conditions.** Spectral ratios of surface albedo at $1280\,\mathrm{nm}$ and $1100\,\mathrm{nm}$ wavelength were used to reduce the retrieval uncertainty. The retrieval was applied to the ground-based and airborne observations and validated against optical in situ observations of SSA utilizing an IceCube device. The SSA retrieved from CORAS observations varied between $27\,\mathrm{m^2\,kg^{-1}}$ and $89\,\mathrm{m^2\,kg^{-1}}$. Snowfall events caused distinct
25  relative maxima of the SSA which were followed by a gradual decrease in SSA due to snow metamorphism and wind-induced transport of fresh fallen ice crystals. **The ability of the modified algorithm to include measurements in overcast conditions improved the data coverage especially at times when precipitation events occured and the SSA changed quickly.** SSA retrieved from measurements with CORAS and MODIS agree with the in situ observations within the ranges given by the measurement uncertainties. However, SSA retrieved from the airborne SMART data underestimated the ground-based results
30  by a factor of 2.

P2 L3:

'most of the sea ice are covered with snow with little seasonal variability'. The variability of sea ice is huge in Antarctica.

Thanks for pointing out this potential misconception. We did not mean the variability of sea ice, but of the snow cover on the sea ice. We reformulated the sentence.

**Changes in manuscript:**

In Antarctica, more than 99.8 % of the continent and most of the sea ice are covered with snow **all year round** (Burton-Johnson
5    et al., 2016).

P2 L9:

'Furthermore, snow surface albedo varies with snow depth and the liquid water content'. Need a reference for each effect. The dependence to snow depth is almost irrelevant in Antarctica.

Thank you. We added exemplary references for the two parameters influencing the snow surface albedo. The importance of the dependence on snow depth for the study is indeed small, however, we mention this parameter for comprehension.

**Changes in manuscript:**

**It further depends** on wavelength (e.g., Hudson et al., 2006; Warren and Brandt, 2008) **as well as** solar position (e.g., Wis-
10    combe and Warren, 1980; Wiscombe, 1980; Dumont et al., 2010) and varies with snow depth **(e.g., Wiscombe and Warren, 1980)** and liquid water content **(e.g., Wiscombe and Warren, 1980; Gallet et al., 2014)**.

P2 L11:

Add references of published works (Libois et al., Munneke et al., Picard et al., Pirrazini et al., Warren and co, Zender and co, etc).

We added some additional references to demonstrate the influence of snow grain size on snow albedo.

**Changes in manuscript:**

From these parameters, the snow grain size has the largest effect on snow albedo. **Simulations by Wiscombe and Warren**
15    **(1980) showed that the spectral albedo in the near-infrared part of the spectrum may drop by a factor of 2 or more when the snow grain size increases from 50 µm to 1000 µm.** Dang et al. (2015) showed that the transition from fresh fallen snow with a typical effective grain size of 100 µm to **aged** snow (1000 µm) leads to a decrease in snow albedo (spectrally integrated over 0.3-4.0 µm) of 15 % from 0.83 to 0.72. **This relation could clearly be identified in ground-based measurements by Domine et al. (2006) analyzing snow samples taken at Svalbard in 2001. They measured a decrease in reflectance at**
20    **1310 nm wavelength by 45 % with increasing grain size from 290 µm to 1175 µm. Libois et al. (2015) and Picard et al. (2016) retrieved a three-year time series of SSA from spectral albedo measurements at Dome C, Antarctica, which**

emphasized the dynamical evolution of near-surface SSA by displaying a 3-fold decrease every summer in response to the increase of air temperature. A detailed comparison of this dataset with snow model simulations and a geophysical interpretation are presented by Libois et al. (2015). Similar seasonal variations but little year-to-year variation was found by Jin et al. (2008) who retrieved snow grain size from measurements with the MODerate Resolution Imaging

5 Spectroradiometer (MODIS) at 1.64 and 0.64 µm over the Antarctic continent for four days each year between 2000 and 2005. Therefore, a positive feedback mechanism can be postulated: increasing snow temperatures are followed by an accelerated snow metamorphism and a decrease of the snow albedo, which leads to higher absorption and heating of the snow. However, the expected snow albedo decrease due to temperature-induced metamorphism (0.3 % for a warming of 3 K) can be overcompensated by an increase in snow albedo by 0.4 % owing to a projected increase in precipitation

10 during the twenty-first century in interior Antarctica (Picard et al.,2012).
* * *
P2 L18:

Are these values here for optical grain size ? Is the reference about optical grain size or other grain size metrics ? The sentence should be more precise about this.
* * *
The reference is about the geometric grain size (visually determined by means of a magnifying glass). We added this information in the manuscript.

**Changes in manuscript:**

Observations showed that the **geometric** grain size **(from visual determination)** of snow crystals varies between 10 µm for

15 fresh fallen snow and up to 3 mm for **aged** snow (Singh, 2001).
* * *
P3 L7:

It is not clear if the sentence is about the impact of the albedo-grain shape or BRDF-grain shape. Both are quite different and have different impact depending on the algorithm used for the retrieval. For the former issue, the value of 2.6 is extreme. See Kokhanovsky and Zege 2004, Picard et al. 2009, Libois et al. 2013 for other works on this question.
* * *
The study of Dang et al. (2016) deals with the influence of grain shape on the snow albedo, not the BRDF. Among other things, Dang et al. (2016) simulate the spectral albedo at 1300 nm wavelength for non-spherical snow grains (with aspect ratio 1) and spherical snow grains. A measured albedo of 0.59 would cause a retrieved effective radius of 60 µm (spherical) or 144 µm (nonspherical). Therefore, the representation of non-spherical snow grains by a population of spherical grains with the same

20 area-to-mass ratio can lead to an underestimation of the effective snow grain size by a factor of 2.4.

Thank you for the suggested references. We want to take this opportunity and included a second reference on this question.

**Changes in manuscript:**

These albedo models mostly assume spherical grains, which is **unrealistic** because the grain shape is usually far from being spherical **(e.g., Kokhanovsky and Zege, 2004; Libois et al., 2013; Leppänen et al., 2015). Picard et al. (2009) estimated an**

**uncertainty of $\pm 20\,\%$ when determining SSA from albedo measurements in case of an unknown snow grain shape. A common approach to account for the non-spherical shape of snow grains is to represent the non-spherical snow grains by a population of spherical grains with the same area-to-mass ratio in the spectral albedo model. However,** as shown by Dang et al. (2016), **this approximation** can lead to an underestimation of the **retrieved** effective snow grain size by a factor

5 of **more than 2**.

About P3, Libois et al. 2015 and Picard et al. 2016 (both in The Cryosphere), have produced time-series of SSA (3 years) at Dome C in Antarctica which seems relevant to the present study. See also Jin et al. 2008 and Picard et al. 2012 for earlier (and shorter) time-series of SSA in-situ and satellites measurements.

This is true, thank you. We added the suggested studies to the introduction.

**Changes in manuscript:**

See comment above about P2L11.

P3 L12-13:

'However, polar orbiting satellites do not provide a sufficiently high temporal resolution that may resolve snow precipitation events and snow metamorphism that typically can advance in a matter of hours'. With the number of satellites (MODIS, Sentinel, SSM/I, AMSU, ...) and the convergence of the orbits in the polar regions, subdaily resolution can be achieved.

This is true, in theory a subdaily resolution could be achieved. However, to our knowledge no hourly satellite products of snow

10 grain size are currently available. We rephrased the sentence in order to make it less definitive.

**Changes in manuscript:**

**Up to now, grain size products of** polar orbiting satellites do not provide a sufficiently high temporal resolution that may resolve snow precipitation and metamorphism that typically can advance in a matter of hours.

P4L8:

According to Gallet et al. 2009, IceCUBE (i.e. DIFUSSS with 1300 nm only) can not be used from SSA of 60m2/kg and above. If the "in prep" paper shows different results, it would be interesting to add a sentence here. Otherwise the limitation should be indicated.

Thank you for indicating that this limitation was not yet stated in the manuscript. However, the snow samples for which Gallet

15 et al. (2009) reported problems with SSA of $60\,\mathrm{m^2\,kg^{-1}}$ and above were all of low density below $100\,\mathrm{kg\,m^{-3}}$. Within this study, the density of the snow samples used for the SSA measurements were all well above $100\,\mathrm{kg\,m^{-3}}$ (around 60 % of the samples with densities between 280-350 $\mathrm{kg\,m^{-3}}$). The higher optical depth of the samples than in Gallet et al. (2009) might indicate a higher limit for the SSA measurements. Still, we included in the manuscript that SSA values above $60\,\mathrm{m^2\,kg^{-1}}$

require caution. Out of a total of 4900 single measurements, SSA values above $60\,\mathrm{m^2\,kg^{-1}}$ were recorded 499 times on 17 different days (10 %).

**Changes in manuscript:**

**SSA values above $60\,\mathrm{m^2\,kg^{-1}}$ require caution as the insufficient optical depth of the snow sample may cause artifacts**
5 **as reported by Gallet et al. (2009). However, the densities of the snow samples for which Gallet et al. (2009) reported this limitation were below $100\,\mathrm{kg\,m^{-3}}$, whereas the observed snow densities within this study were all well above this value (around 60 % of the samples with densities between 280-350 $\mathrm{kg\,m^{-3}}$). The higher optical depth of the samples might indicate a higher limit for the SSA measurements. However, SSA values above $60\,\mathrm{m^2\,kg^{-1}}$ occured only in 10 % of the measurements..**

> P4 L15:
>
> invert the temperatures

10 We followed your suggestion and changed it in the manuscript.

> Fig 1:
>
> I'm not convinced that this first figure (the octa plot) is useful for the paper. It shows raw data that are not used as is in the following and the interpretation is short and inconclusive. Instead, I suggest to plot a time-series (a normal one as for the air temperature, not a time versus date graph) cloudy/not cloudy as used by the algorithm to switch between diffuse/direct radiation.

Thank you for your opinion and your suggestion for improvement. However, we think the hourly cloud cover plot is helpful to the reader in the way it is presented in the manuscript. Changing to a single time series of cloudy/cloudfree looks too crowded for the two months of measurements as cloudfree/cloudy periods partly switch too often. In the hourly plot, the times when measurements are analyzed can be identified much more precisely. We, therefore, added open and filled circles (corresponding
15 to the same symbols used in Fig. 6) to the octa plot in order to both show the exact time of retrieval and whether the algorithm uses cloudless or overcast conditions. In that way, the cloud cover plot is much more meaningful compared to the original version. We adjusted the manuscript accordingly to account for the changes in Fig. 1.

**Changes in manuscript:**

[Figure]

**Figure 1.** Time series of 2 m air temperature (red line) and hourly cloud cover (blueish squares) at Kohnen station between 10 December 2013 and 31 January 2014. Snowflake symbols denote days with snowfall. **Black circles show times when SSA was retrieved from CORAS measurements (Fig. 6) and denote retrievals for cloudless (open circles) and overcast (filled circles) conditions.**

**3.2.1 Application in cloudless conditions**
**For each day, the times when SSA was retrieved from measurements with CORAS are added to Fig. 1 as black open circles. It has to be noted that the cloud cover was estimated from visual observation every full hour, whereas CORAS measurements were partly analyzed for times in between the visual observation and the cloud cover given here might**

5 **not be representative for the actual CORAS measurement (e.g., 5 octa on 27 December 2013). Therefore, the CORAS measurements were carefully screened for any cloud contamination by analyzing the downward solar irradiance to guarantee homogeneous cloudless or overcast conditions.**
**3.2.2 Extension to overcast conditions**
**Retrievals in overcast conditions were applied on four days and are denoted as black filled circles in Fig. 1.**

P5L1:

'polar day'. This term is ambiguous and the sentence is not strictly logical, air temperature decrease observed before the end of the 'polar day' is due to a change of solar elevation, not a change of day duration.

10 Thank you for pointing this out. We followed your suggestion and changed it in the manuscript.
**Changes in manuscript:**
Towards the end of the measurement period, the temperature level decreased due to **lower sun elevations**.

P5L10:

where these values of uncertainty are coming from ? As explained in the general comment, the description of the CORAS instrument and the evaluation of the uncertainty are missing. This part needs to be expanded but at least one paragraph. Please also consider the shadowing effect, frost formation, ... Information such as the cosine correction method, the frequency of observation, and other instrumental details are needed as well.

We carefully revised the uncertainty estimation and its description to make clear where the uncertainties of each measurement result from.

**Changes in manuscript:**

The spectral resolution is 2 to 3 nm between 0.3 and 1.0 µm and 15 nm up to 2.2 µm wavelength, **with a full spectrum mea-**
5  **sured every 15 s. The uncertainties of albedo measurements with CORAS range between 4.0 to 8.0 % depending on wavelength and combining different sources of instrumental errors. The signal-to-noise ratio accounts for 1.3-3.0 % uncertainty. Dark spectra were recorded constantly throughout the measurements resulting in a reliable correction for dark current and stray light within the spectrometer (0.1 % uncertainty). The wavelength calibration of the spectrometer accounts for 1.0 % uncertainty. For albedo measurements, the two optical inlets were cross-calibrated with an**
10  **identical radiation source at four times during the observation period. The temporal stability of this cross-calibration is estimated with 1.0-4.5 % depending on wavelength. Furthermore, the non-ideal cosine characteristics of the irradiance optical inlets were characterized within the laboratory. The optical inlets were mounted on a turn table and the lamp signal under different angles of incidence (-95° to +95° in steps of 5°) was recorded. This procedure was repeated for four different relative azimuth angles between lamp and optical inlet and was used to compute correction factors for**
15  **the cosine response depending on solar zenith angle, wavelength, and the direct fraction of the global irradiance. The azimuthal stability of the correction factors of 3.5 % was used to estimate the instrumental errors attributed to the non-ideal cosine response of the optical inlets. This way, the instrumental uncertainties combine to 6.8 % at 1280 nm wavelength. The analyzed measurement times were carefully selected to avoid errors due to frost formation and shadow effects which were typically observed during early morning.**
20  (...)

SMART **albedo** measurements have an estimated uncertainty between **4.1 to 8.1 %** also taking into consideration uncertainties due to the active horizontal stabilization of 1.0 %.

(...)

**The measurement uncertainty of albedo measurements at 1280 nm wavelength was estimated with 6.8 %.** Using $\mathcal{R}$, the
25  uncertainty is reduced to **5.5 %** as the transition to relative measurements yields independence from the **cross-calibration. In addition to the instrumental errors, the surface slope at the footprint scale may influence the retrieval results. Using data from Picard et al. (2016), Dumont et al. (2017) found variations of the surface slope caused by wind drift at Dome C of $\pm$ 2°. They estimated a resulting uncertainty of 10 % in retrieved SSA due to these variations of the surface slope. Assuming a similar variability of the slope of the surface at Kohnen, an additional uncertainty of 10 % is assumed**

**for the retrieval of SSA.** Applying Eq. 12 to the spectral albedo measured by CORAS on 24 December 2013 (Fig. 4a), the measured ratio $\mathcal{R}$ of **0.702$\pm$0.039** would lead to an estimated effective snow grain size of about **90$\pm$31 $\mu$m** at $\theta_0 = 54°$ (blue and red lines in Fig. 4b). However, the relative uncertainty of the retrieval varies with solar position and effective snow grain size. In general, it is higher for smaller snow grains. Overall, the retrieval uncertainty ranges between **25 % and 37 %** for the

5    SSA throughout the measurement period.

(...)

The uncertainties of the retrieval were estimated similarly to the ground-based CORAS measurements, with the exception that the uncertainty of irradiance measurements is assumed to be slightly higher due to the remaining uncertainty in the horizontal leveling of the airborne sensors by the horizontal stabilization of SMART **(6.9 % at 1280 nm wavelength)**. As a result, the

10    estimated uncertainty of the measured albedo ratio $\mathcal{R}$ is about **5.6 %.**

P7L20:

Regarding the choice of the values of B and g in the 'middle of the range':

1) Values obtained from measurements are now available. Libois et al. 2013 and 2014 have measured B and recommended some values of g based on measurements.

2) The uncertainty of these values impacts the SSA estimation and needs to be taken into account in the evaluation of the uncertainties proposed in this paper which only considers instrumental uncertainties.

Thank you for this comment! Indeed, values of B and g based on measurements have been reported by Libois et al. The general recommendation by these authors is to refrain from spherical snow grains and use the shapes with B=1.6$\pm$0.2 to model snow grains. In our approach, we use B=1.5 which is in the range recommended by Libois et al. We are well aware that an inaccuracy in the particle form factor will affect the snow grain size retrieval. According to Zege et al. (2011), the retrieval inaccuracy

15    which may stem from the wrong assumption of A is less than 25 %. This has been highlighted in the text of the manuscript.

**Changes in manuscript:**

Within the SGSP algorithm, a value of 5.8 is used as an average value over a mixture of randomly oriented hexagonal plates and columns with $B(\xi) \approx 1.5$ and $g(\xi) \approx 0.84$. **This is in accordance with Libois et al. (2014) who recommend a value for** $B(\xi)$ **within 1.6 $\pm$ 0.2 based on measurements in Antarctica and the French Alps. Any inaccuracy in the form factor** $A$

20    **will affect the snow grain size retrieval in addition to instrumental errors. According to Zege et al. (2011), the retrieval inaccuracy which may stem from a false assumption of** $A$ **is less than 25 %.**

P8L5-L8:

A few critical information are missing about the MODIS algorithm. - Which BRDF parameterization has been used and how ? - How the radiances have been converted to ground-level surface reflectances ? - Has an atmospheric model been used ? - Which product and version have been used here ?

The following clarifications are included into the manuscript:

1. MODIS Level 1B Collection 5 data have been used.

2. The angular reflective properties of the medium have been set via the particle form factor and the photon escape function. The BRDF is assumed to be wavelength independent.

3. The effect of the atmosphere has been removed using the radiative transfer model RAY (Tynes et al., 2011).

4. Subarctic winter atmospheric model (Kneizys et al., 1996) and Arctic background aerosol model (Tomasi et al., 2007) were used for constant atmospheric conditions. The amount of aerosol in Antarctica is in general two times less than in the Arctic (aerosol optical thickness 0.05 in the Arctic and 0.02-0.03 in the Antarctic at 500nm, from Tomasi et al. 2007 and AERONET data). The effect of this very low pollution onto the retrieval is considered to be negligible at the retrieval channels.

**Changes in manuscript:**

For the retrieval of effective snow grain size from satellite data, snow-atmosphere radiative interactions have to be taken into account by employing an atmosphere model as described in Zege et al. (2011). **The effect of the atmosphere has been removed employing the radiative transfer model RAY (Tynes et al., 2001) using the subarctic winter atmospheric model (Kneizys et al., 1996) and the Arctic background aerosol model (Tomasi et al., 2007) for constant atmospheric conditions. The effect of the very low pollution in Antarctica (e.g., AOD of 0.015 at 500 nm at Kohnen station, Tomasi et al., 2007) onto the retrieval is considered to be negligible at the retrieval channels.**

Radiance data from MODIS (**Level 1B Collection 5**) onboard the Aqua and Terra satellites were used to retrieve effective snow grain sizes in the entire area covered by the campaign. The SGSP algorithm was applied for areas identified as cloudless. After a preliminary separation of snow pixels, the effective snow grain size of each pixel is retrieved from radiance measurements of MODIS channels 3 (469 nm wavelength), 2 (858 nm), and 5 (1240 nm). **Assuming a spectrally constant bidirectional reflectance distribution function (BRDF) of snow and using the combination of three spectral channels, the angular dependency of the measured radiance is excluded.**

P9L13:

The argument implicitly assumes that the calibration error is wavelength-independent, which is often not the case. Even if the ratio R is less sensitive to calibration issues, the residual wavelength dependence can greatly affects the estimation of SSA as shown in Picard et al. 2016 (e.g. cosine correction, 100% direct assumption...).

We carefully revised the section on the measurement uncertainties and extended the description of the various sources of uncertainty. For the changes made in the manuscript, please see the comment above.

P8L16:

these values have been obtained for perfect clear-sky. To my personal experience, direct/diffuse ratio shows huge variations in the infrared due to thin cloud (more than in the visible because the diffuse component is already quite large). Calculations of the atmosphere model with cloud with varying optical depth and crystal size would be useful, because the estimation of SSA using ratio (i.e. Eq 9) is sensitive to the spurious wavelength-dependence resulting from cloud cover.

It is true that thin clouds would change the direct/diffuse ratio significantly. We applied the retrieval under cloudless conditions using Eq. 9 only after a careful manual selection of the retrieval times based on a comparison between measured and simulated downward irradiance. Therefore, it was assured that no thin clouds did increase the diffuse radiation and contaminated the measurements and Eq. 3 is applicable. Nevertheless, also in cloudless conditions, the diffuse radiation is not zero. From

5  radiative transfer simulations, we estimated the fraction of direct solar radiation with respect to the global irradiance to lie between 94.6 and 99.8 %. This value was used to quantify the retrieval uncertainties due to the assumption of 100 % direct radiation. However, we think a detailed sensitivity study with respect to cloud optical depth and crystal size would be beyond the scope of this study.

**Changes in manuscript:**

10  Hence, **after careful selection of cloudless periods,** the effective snow grain size can be calculated directly by inverting Eqs. 3 and 8.

P10L7:

I suggest to improve a little the justification of the constance of SSA during the day. E.g. doi:10.5194/tc-8-1205-2014

Thank you very much for pointing out this reference. Gallet et al. (2014) indeed found diurnal variations in SSA due to crystal growth in the same order of magnitude as observed from the CORAS retrieval. However, the changes observed by Gallet et al. strongly depend on meteorological conditions. Therefore, variations due to crystal growth can be not symmetric to noon and

15  even change the sign on a day-to-day scale. Although an influence of crystal growth on the SSA variations cannot be ruled out for the measurements presented here, the evident dependence on the solar zenith angle and the symmetry to noon of the observed diurnal cycle of the SSA retrieved from CORAS strongly indicate that the crystal growth influences the results less than the changing solar zenith angle.

**Changes in manuscript:**

20  **Furthermore, Gallet et al. (2014b) observed SSA variations on a sub-daily scale using the DUFISSS instrument (DUal Frequency Integrating Sphere for Snow SSA measurements; Gallet et al., 2009) at Dome C in January 2009. They measured a drop in SSA at around noon from $40\,\mathrm{m^2\,kg^{-1}}$ to $33\,\mathrm{m^2\,kg^{-1}}$ before the SSA increased again to $41\,\mathrm{m^2\,kg^{-1}}$ at midnight. These temporal changes were attributed to the growth of sublimation crystals during daytime and nighttime formation of surface hoar. The SSA variations observed by CORAS on 24 December 2013 are in the same order of mag-**

25  **nitude. However, the variations observed by Gallet et al. (2014b) are not symmetric to noon and their sign changes on a day-to-day scale due to the strong dependence on meteorological conditions. Even though an influence of crystal growth**

**processes cannot be ruled out for the measurements presented here, the evident dependence on the solar zenith angle and the constant symmetry to noon of the diurnal cycle observed in the SSA retrieved from CORAS measurements strongly indicate a dominating influence of the solar zenith angle.**

P10L10:

the surface roughness/slope at the footprint scale is another likely player that affects the theory used here. See e.g. Dumont et al. 2017 in TC (see the slope estimation).

Thank you for pointing at this specific reference. We included a discussion on the uncertainties introduced by the unknown slope at the footprint scale (see comment above).

P14L5:

the argument on the scales seems weak because the airborne data are between the in-situ and MODIS data in terms of scales and MODIS shows little bias w/r to in-situ measurements. I guess that the authors have analyzed in details the spectra and various sources of errors from the different sensors without success. May be add part of this analysis. For instance, showing coincident spectra from the different sensors could help to understand the differences, and at least to rule out some potential defects of the sensors.

It is true that the airborne data are between the in situ and MODIS data in terms of scales. However, whereas MODIS averages over $50 \times 50 \, \text{km}^2$ with Kohnen station situated in the central pixel, 
[revised manuscript text omitted]
_{\text{ext}} = \frac{1.5\,C_{\text{v}}}{r_{\text{eff}}} \qquad \text{and} \qquad b_{\text{abs}} = B(\xi) \cdot b_{\text{abs,ice}} \cdot C_{\text{v}} = B(\xi) \cdot \frac{4\pi\chi}{\lambda} \cdot C_{\text{v}},$$

$$b_{\text{ext}} = \frac{1.5\,C_{\text{v}}}{r_{\text{eff}}}, \qquad \text{and} \tag{6}$$

$$b_{\text{abs}} = B(\xi) \cdot b_{\text{abs,ice}} \cdot C_{\text{v}} = B(\xi) \cdot \frac{4\pi\chi}{\lambda} \cdot C_{\text{v}}. \tag{7}$$

$b_{\text{ext}}$ and $b_{\text{abs}}$ only depend on the volumetric concentration of snow grains $C_{\text{v}}$ (dimensionless, $\rho_{\text{snow}}/\rho_{\text{ice}}$), the effective snow grain size $r_{\text{
[revised manuscript text omitted]

---

## Author Response (AR2)

**Reply to Referee reviews**

**Manuscript title**

Comparison of different methods to retrieve optical-equivalent snow grain size in central Antarctica

**Date of reply**

27 September 2017

Dear Editor and Reviewer,

thank you for the helpful comments and suggestions on improving the manuscript.

The detailed replies to your comments (highlighted in gray) are given below, followed by the specific changes made in the manuscript (changes are highlighted in bold text). The marked-up version (see below) illustrates all changes made compared

10 to the last version.

We recently corrected for an inconsistency in the data analysis of the airborne measurements. This compensates for most of the underestimation of the SSA retrieved by SMART compared to the in situ measurements (factor of 1.2 instead of 2.1). We modified Figure 6 accordingly and updated parts of the discussion concerning this underestimation. The specific changes are illustrated in the marked-up version below.

15 We published the time series of SSA retrieved from CORAS, SMART, and MODIS (from Fig. 6) in PANGAEA and included a reference to the doi of this dataset at the end of the manuscript (**Data availability**).

Kind regards from the authors.

**Replies to comments of Referee 3 (H. Löwe)**

I would only make "grain size terms" consistent. This point has been raised before and was indeed addressed in the revision. In principle the "fix" taken by the authors in the body of the text is acceptable (maybe not the most elegant), but it should be made consistent in the following way:
l14, p2: delete "effective"

Changed as suggested.

l32, p2: delete "geometric"

Changed as suggested.

l1, p3: rephrase "The traditional grain size is defined by...."

Changed as suggested.

**Changes in manuscript:**
The **traditional** grain size is **defined by the length** of the largest extension of a snow grain (...)

p3, l2: rephrase "Therefore not only for radiative transfer applications an effective grain size $r_{eff}$ is defined as the optical grain size, i.e. the optical equivalent grain radius of a collection of spheres with..."
Then the first part of the introduction only uses "grain size" in the loose (container term) sense. Then it comes to the definitions, where "traditional grain size" is contrasted to "optical grain size" and related to "SSA". In addition its made clear that in this paper "effective grain size" is actually a synonym for "optical grain size". The abstract is fine in this respect, but i would also avoid "effective" in the title, just use "optical grain size". Then everybody knows immediately what's going on.
BTW: Why in fact going the detour of defining an "effective grain size" which is in the end nothing but the the optical grain size? So a an even better polish would be to simply avoid "effective grain size" at all and replace it by "optical grain size" throughout... Anyway.

Thank you for your efforts concerning a consistent terminology within this paper. We followed and applied your last comment. In doing so, we carefully revised the text and changed 'effective' to 'optical'. In addition, we modified formulas, figures (Fig. 4, 5, and 6), and the title accordingly. The new title reads: Comparison of different methods to retrieve **optical-equivalent** snow grain size in central Antarctica.

> At some point the authors may give (Krol, Löwe, TC, 10, 2847?2863, 2016) a read in view of grain shape.

Thanks for pointing at this publication. We included a reference on Page 3 (line 21) as it further underlines the influence of the grain shape.

**Replies to comments of Editor (C. Fierz)**

Sorting out grain size, traditional grain size, optical grain size, and effective grain size is a tedious task indeed. We went back and forth but I think we are on the good way if we follow Henning Löwe's last recommendation under B) in his review, 'So an even better polish would be to simply avoid "effective grain size" at all and replace it by "optical grain size" throughout... Anyway.'

I think this approach also fits best Ghislain Picard's recommendations and can really be applied throughout the paper (see attached).

We agree and carefully applied the necessary changes to the manuscript (see also reply on referee comment above). The specific changes made in the manuscript can be found in the marked-up version below.

In addition, I made in the attached a few suggestions that I'd ask you to consider.

Thank you for your suggestions, we applied the following changes based on your comments:

- We changed the title: Comparison of different methods to retrieve **optical-equivalent** snow grain size in central Antarctica.

- We rephrased the first sentences of the abstract to make the statement more rigorous.
  The **optical-equivalent snow grain size** affects the reflectivity of snow surfaces and, thus, the local surface energy budget in particular in polar regions. Therefore, the specific surface area (SSA), **from which the optical snow grain size is derived**, was observed for a two-month period in central Antarctica (Kohnen research station) during austral summer 2013/14.

- p.2 l.32: The stated $10\,\mu m$ for fresh fallen snow are given in Singh (2001) as a lower boundary for the traditional grain size of new snow (0.01 - 0.5 mm).

- p.2 l.34: Rephrased the sentence as suggested: **Note that this snow metamorphism is also** effective below the freezing temperature. **It mainly depends on snow microstructure, snow** temperature, and **its vertical gradient within the snowpack.**

- about p.2: We left the order of the two paragraphs unchanged as we think the spectral absorption of snow grains needs to be explained before the optical SSA measurements are listed. In addition, we think that in leaving out 'geometric', the citation of Singh (2001) does not necessarily need an earlier definition of the traditional grain size.

[revised manuscript text omitted]

---

## Author Response (AR3)

**Reply to Editor comment**

**Manuscript title**

Comparison of different methods to retrieve optical-equivalent snow grain size in central Antarctica

**Date of reply**

10 October 2017

Dear Editor,

The detailed reply to your comment (highlighted in gray) is given below, followed by the specific changes made in the manuscript (changes are highlighted in bold text). The marked-up version (see below) illustrates all changes made compared to the last version.

10 In addition, we removed a citation on p.2 line 11 (Wiscombe, 1980) as it was still a remnant from an earlier version of the manuscript and revised the citation on p.3 line 16 which was supposed to be (Wiscombe and Warren, 1980) instead of (Wiscombe, 1980).

Kind regards from the authors and thanks for your contribution.

**Reply to comment by Editor (C. Fierz)**

However and unfortunately, I still have an issue with the sentence on lines 33-34, p2: "Observations showed that the grain size (from visual determination) of snow crystals varies between 10 μm for fresh fallen snow and up to 3 mm for aged snow (Singh, 2001)."

Indeed, somehow the sentence does not make sense to me - and I think to many snow scientists. Moreover, table 3.3 on page 108 of Singh (2001) is far from trustworthy in my view: where do this numbers come from? What is 'old snow'? etc. Thus I would suggest to change this to either:

A) "It is well known from visual determination that the size of snow crystals and grains may vary from 0.05 up to at least 10 mm (for example, see Jordan et al., 2008 and Pomeroy and Brun, 2002)."?In my view, this sentence could also be left without referencing anybody as it is common sense!

B) "It is well known from visual determination that snow crystals reaching the ground typically range by two order of magnitudes in their maximum dimensions from 0.050 to 5 mm (Jordan et al., 2008, p.17) while depth hoar crystals can reach sizes of 10 mm or more (Pomeroy and Brun, 2002, p. 99)."

or C) if you want to reference work that is more specific to Antarctica, see Gay et al. (2002 ) below.

We followed your suggestion A and left the revised sentence without reference.

**Changes in the manuscript:**

[revised manuscript text omitted]